# Smartphone app for non-invasive detection of anemia using only patient-sourced photos

Robert G. Mannino[1,2,3], David R. Myers[1,2,3], Erika A. Tyburski[1,2,3], Christina Caruso[2], Jeanne Boudreaux[2], Traci Leong[4], G. D. Clifford[1,5] & Wilbur A. Lam[1,2,3]

We introduce a paradigm of completely non-invasive, on-demand diagnostics that may replace common blood-based laboratory tests using only a smartphone app and photos. We initially targeted anemia, a blood condition characterized by low blood hemoglobin levels that afflicts >2 billion people. Our app estimates hemoglobin levels by analyzing color and metadata of fingernail bed smartphone photos and detects anemia (hemoglobin levels <12.5 g dL$^{-1}$) with an accuracy of ±2.4 g dL$^{-1}$ and a sensitivity of 97% (95% CI, 89–100%) when compared with CBC hemoglobin levels ($n = 100$ subjects), indicating its viability to serve as a non-invasive anemia screening tool. Moreover, with personalized calibration, this system achieves an accuracy of ±0.92 g dL$^{-1}$ of CBC hemoglobin levels ($n = 16$), empowering chronic anemia patients to serially monitor their hemoglobin levels instantaneously and remotely. Our on-demand system enables anyone with a smartphone to download an app and immediately detect anemia anywhere and anytime.

[1] The Wallace H. Coulter Department of Biomedical Engineering at Georgia Tech and Emory University, Atlanta, GA, USA. [2] Aflac Cancer and Blood Disorders Center of Children's Healthcare of Atlanta and Department of Pediatrics, Emory University School of Medicine, Atlanta, GA, USA. [3] The Parker H. Petit Institute for Bioengineering and Biosciences, Georgia Institute of Technology, Atlanta, GA, USA. [4] Department of Biostatistics & Bioinformatics, Rollins School of Public Health, Emory University, Atlanta, GA, USA. [5] Department of Biomedical Informatics, Emory University, Atlanta, GA, USA. Correspondence and requests for materials should be addressed to W.A.L. (email: wilbur.lam@emory.edu)

Although smartphone-based telehealth has the potential to change how healthcare is delivered by enabling remote diagnosis[1], these technologies have yet to non-invasively replace blood-based testing, which remains a major cornerstone of disease diagnosis in modern medicine[2–4]. Indeed, while custom-designed smartphone attachments may allow for blood-based, point-of-care diagnosis and analysis[5], the burden of blood sampling combined with the additional cost and need for equipment external to the smartphone prevents broader adoption of these potentially disruptive technologies.

Due to its high prevalence affecting over 2 billion people globally, anemia, characterized by low blood hemoglobin (Hgb) levels[6], was chosen as the initial disease target for us to study. Anemia has numerous causes, ranging from common nutritional causes, such as iron or folate deficiency, which are relatively straightforward to treat and cure, to rarer genetic causes, such as sickle cell disease or thalassemia major, which lead to severe and chronic anemia that requires frequent monitoring. Detection of anemia involves either anemia screening or anemia diagnosis, and both require different degrees of measurement accuracy. First, a clear clinical need exists for easily and widely accessible tools to screen for anemia among at-risk populations (e.g., pregnant women, toddler-age children, elderly patients) or the general public to determine whether an individual will need formal confirmatory testing with the gold standard Hgb level test obtained via a complete blood count (CBC). However, there is also a need to for non-invasive methods to more quantitatively and officially diagnose and monitor anemia with higher precision Hgb levels, especially patients with known or chronic anemia.

Existing clinical approaches to measure blood Hgb levels require specialized equipment and represent tradeoffs between invasiveness, accuracy, infrastructure requirements, and cost[7–9], all of which are especially problematic in rural and low-resource settings, where anemia is most prevalent. For example, the gold standard CBC Hgb level measurement requires: blood sampling by a trained phlebotomist, a clinical hematology analyzer with the required electrical power, biochemical reagents, and infrastructure thereof, along with a trained laboratory technician to perform the analysis[10]. Aside from being cost-prohibitive in resource-poor settings, the necessary invasive blood sampling to measure Hgb levels causes discomfort in younger pediatric patients[11]. Interestingly, several reports have suggested that anemia may qualitatively correlate with subjective assessments of pallor in various anatomic regions of the patient's body[12–17], namely the fingernail beds, conjunctiva, and palmar creases.

Here, we leverage this observation that pallor is associated with anemia to develop a method that quantitatively analyzes pallor in patient-sourced photos using image analysis algorithms to enable a noninvasive, accurate quantitative smartphone app for detecting anemia (Fig. 1a, Supplementary Movie 1). At the same time, we implemented quality control software to minimize the impact of common fingernail irregularities (e.g., leukonychia and camera flash reflection) on Hgb level measurement (Fig. 1b). To validate our method, we conducted a clinical assessment of this smartphone-based technology using blood samples and smartphone fingernail images of patients with anemia of different etiologies as well as healthy subjects. With this technology, a user downloads an app onto their smartphone, takes a photo of his/her fingernail beds, and instantaneously receives an accurate Hgb level which is displayed directly onto the smartphone screen by the app. Since fingernails, conjunctiva, and palmar creases do not contain melanocytes (melanin producing skin cells), the primary source of color of these anatomical features is blood Hgb[18]. Of these sites, fingernails are straightforward for a user to self-image, unlike conjunctiva, and also have low person-to-person size and shape variability, unlike palmar creases. Our approach represents a substantial conceptual advance over all other published POC anemia detection tools, since these techniques require external equipment, such as calibration cards, background light blocking devices, smartphone attachments, or expensive spectrometry readers[19,20]. Here we have developed a fully functional and standalone smartphone app that enables the non-invasive measurement of blood Hgb levels and has several advantages over existing approaches as highlighted in Table 1. Our app technology leverages the image metadata, a vast trove of information that has been completely ignored by every published study to date that uses smartphones for diagnostics. By mining this rich source of information as well as the color data with a robust multi-linear regression approach, we demonstrate a system in which accurate Hgb measurements are obtained with a smartphone without the need for any external equipment. Indeed, while smartphone images automatically record metadata, instead of examining these data, other groups have used physical strategies such as color calibration cards and light blocking enclosures[20,21]. By eliminating external equipment, this system enables on-demand Hgb level measurement as it requires only the user's smartphone and can be conducted in under 1 min. Therefore, users who desire to screen themselves for anemia can do so immediately by just downloading an app without being required to wait for external equipment to be shipped to their homes, something other smartphone anemia tools require. Furthermore, this smartphone-based technique will empower patients to take control of their clinical care via self-testing of Hgb levels.

## Results

**Anemia screening using the smartphone app.** This system has the capacity to serve as a noninvasive anemia self-screening tool for use by the general population or at risk populations. With a single smartphone image and no personalized calibration step, smartphone Hgb levels were measured to within ± 2.4 g dL$^{-1}$ with a bias of 0.2 g dL$^{-1}$ of CBC Hgb levels in 100 patients with a variety of anemia diagnoses mixed with healthy subjects (Fig. 2a, $r = 0.82$; Fig. 3, Table 2), defined as the 95% limits of agreement (LoA). This noninvasive approach represents a greater degree of accuracy than reported accuracy levels of existing invasive POC anemia screening methods[7,22,23]. Moreover, receiver-operating characteristic analysis revealed that this test achieves a strong diagnostic performance with an area under the curve of 0.88 (Fig. 2b) and highlights the accuracy of this technology throughout the entire range of tested Hgb levels. Additionally, there was minimal correlation between patient Hgb levels and smartphone-measured residual ($r = 0.26$), indicating that the algorithm performance remained consistent throughout range of tested Hgb levels (Fig. 2c). Notably, when using a cutoff of < 11.0 g dL$^{-1}$ to define anemia, a well-established Hgb level threshold (Fig. 2b, Supplementary Figure 1)[12], the sensitivity and specificity of the system to detect anemia was 92% (95% CI, 80–97%) and 76% (95% CI, 62–87%), respectively. Using the average WHO Hgb level cutoff for anemia of 12.5 g dL$^{-1}$ in men and women, the sensitivity of the test improves to 97% (95% CI, 89–100%), indicating the potential for this test to serve as a noninvasive screening tool for anemia (Supplementary Figure 2)[24]. In fact, this degree of accuracy is on par with reported accuracy values in POC settings of the invasive clinically used Hemocue and substantially better than under-development POC screening tools such as HemaApp and conjunctival analysis via photographs[20,21,25].

**Personalized Hgb level measurements using the smartphone app.** Although the accuracy is excellent for a screening tool, we hypothesized that individual calibration of the algorithm could

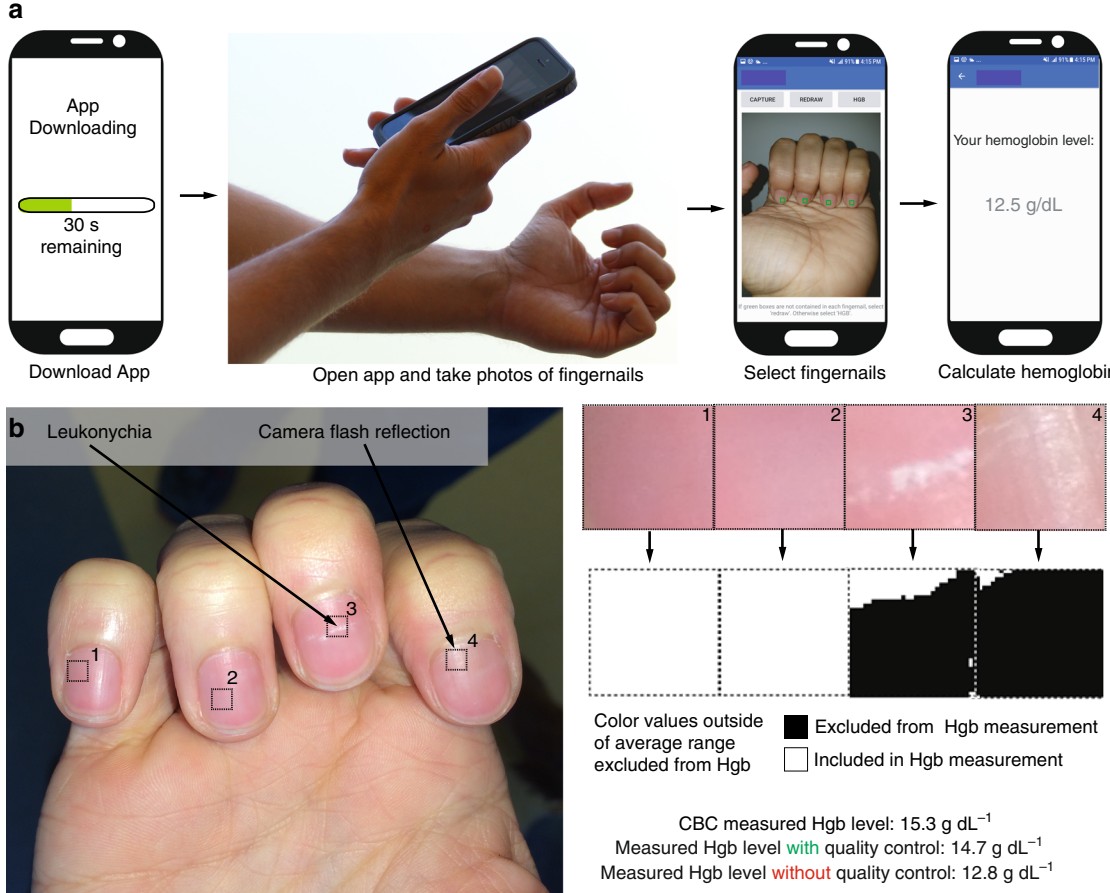

**Fig. 1** Implementation of a smartphone app for measuring hemoglobin (Hgb). **a** A patient simply downloads the app onto their smartphone, opens the app, obtains a smartphone photo of his/her fingernail beds, and without the need for any blood sampling or additional smartphone attachments or external calibration tools, quantitatively measures blood Hgb levels. The patient first takes an image of their fingernails, and is then prompted by the app to tap on the screen to select the regions of interest corresponding to the nailbeds, and a result is then displayed on the smartphone screen. Images are screenshots and photos of actual operation of this app. **b** As smartphone images with fingernail irregularities such as camera flash reflections or leukonychia may affect Hgb level measurements, a quality control algorithm integrated within the Hgb level measurement app detects and omits those irregularities to preserve measurement integrity and accuracy. To that end, the user selects regions of interest from within the fingernail image, and any color values that fall outside of expected color ranges are excluded from Hgb measurement. In this example, when the quality control system was implemented to exclude the fingernail bed irregularities, Hgb level was measured to be 14.7 g dL$^{-1}$, comparable to the patient's CBC Hgb level of 15.3 g dL$^{-1}$. Without the quality control algorithm, Hgb level was measured at 12.8 g dL$^{-1}$, indicating that the algorithm resulted in a 76% reduction in error. Note that as the smartphone image-based algorithm is device-agnostic, the analysis of the smartphone images, and therefore the Hgb level measurements, could also be transmitted to another device (e.g., laptop, cloud-based server) for remote rather than on-board analysis

### Table 1 Comparison of currently existing anemia diagnostic technologies with the app

| Device | Non-invasive | Smartphone-based | Does not require additional equipment (aside from smartphone) | Low cost (<$25) | Accurate (LOA < 3 g dL$^{-1}$) |
|---|---|---|---|---|---|
| Complete blood count hemoglobin levels (Gold Standard) | X | X | X | X | ✓ |
| Hemocue | X | X | X | X | ✓ |
| Masimo co-oximetry | ✓ | X | X | X | X |
| Conjunctival analysis | ✓ | ✓ | X | ✓ | X |
| WHO color scale | X | X | X | ✓ | X |
| HemaApp | ✓ | ✓ | X | ✓ | X |
| Smartphone Anemia App | ✓ | ✓ | ✓ | ✓ | ✓ |

eliminate some of the measurement error introduced by subject-to-subject variability and therefore further improve the accuracy. Thus, we conducted a study in which the smartphone-based algorithm was calibrated for each subject (4 subjects total) over the course of 4 weeks and achieved personalized, accurate Hgb level measurements, enabling long-term serial monitoring of Hgb levels (Fig. 4a, Supplementary Figure 3). Overall, when used in

this manner, this system achieved a level of accuracy of ±0.92 g dL$^{-1}$ with a bias of 0.09 g dL$^{-1}$ compared to CBC Hgb levels (Supplementary Figure 4), again, defined by the 95% LOA (i.e., the Hgb level difference from the gold standard that 95% of smartphone measurements will fall between), representing an improvement on the reported accuracy of current invasive, point-of-care hemoglobin tests, such as Hemocue[9], and clinically used

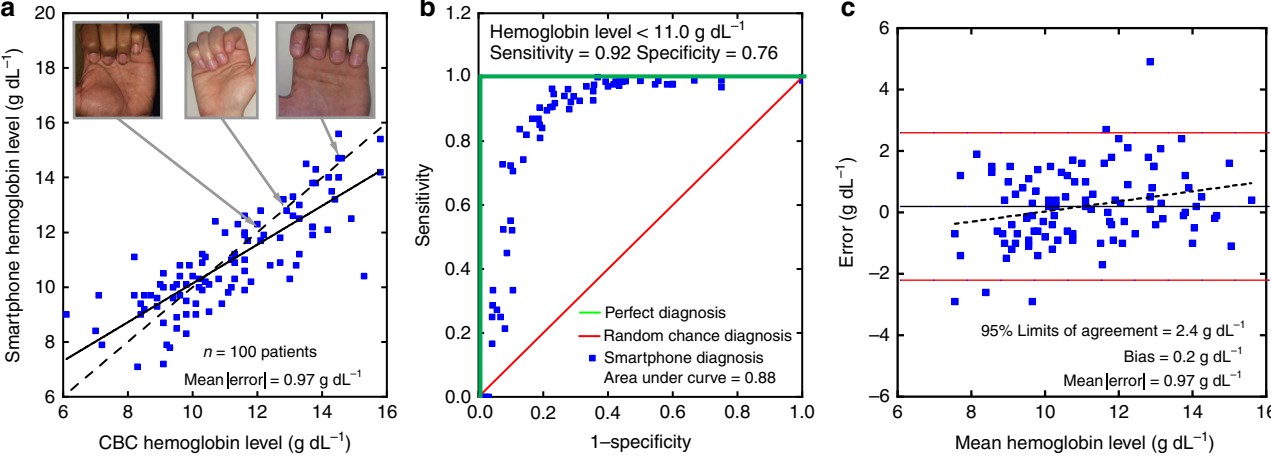

**Fig. 2** The smartphone-based image analysis algorithm accurately measures Hgb levels. **a** The smartphone image analysis algorithm measures blood Hgb levels to within ±0.97 g dL$^{-1}$ of the CBC Hgb level ($r = 0.82$, mean |error|). The solid line represents the ideal result where smartphone Hgb level is equal to the CBC Hgb level whereas the dashed line represents the actual data fit. Inset images illustrate example patient-sourced photos that were used to calculate Hgb level measurements. **b** The receiver-operating characteristic (ROC) analysis graphically illustrates the algorithm's diagnostic performance against a random chance diagnosis (red line), with an area under the curve of 0.5, and a perfect diagnostic (green lines), with an area under the curve of 1. In the case of this noninvasive smartphone app Hgb measurement system (black line), the area under the curve of 0.88 suggests viable diagnostic performance of this algorithm. When using the WHO Hgb level cutoff of < 12.5 g dL$^{-1}$, the sensitivity of the test is 97% (95% CI, 89–100%), $n = 100$ patients. **c** Bland–Altman analysis reveals minimal experimental bias with 0.2 g dL$^{-1}$ average error, indicating that Hgb measurement is has a very small bias. The dashed line represents the relationship between the residual and the average of Hgb level measurements obtained from the CBC and the algorithm ($r = 0.26$). The solid red lines represent 95% limits of agreement (±2.4 g dL$^{-1}$). $n = 100$

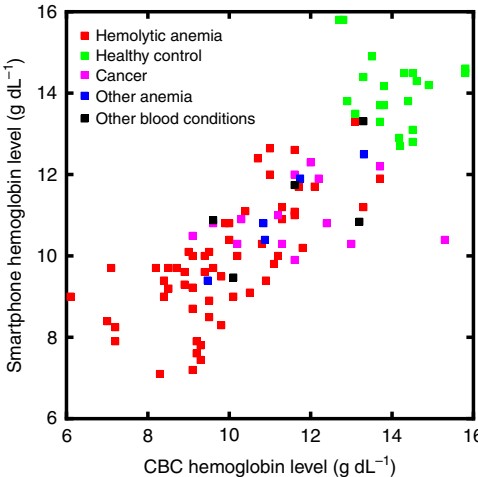

**Fig. 3** Diagnosis profile of our hemoglobin measurement patient population. Subjects with hemolytic anemia, healthy controls, cancer, other anemia (e.g., aplastic anemia), as well as various other blood disorders (e.g., such as thrombocytopenia, deep vein thrombosis, and hemophilia) participated in the study. These data represent the diagnosis profiles of the subjects shown in Fig. 2. $n = 100$

noninvasive methods such as the Masimo Radical 7[26]. The standard deviation used to calculate the 95% LOA in this case was determined via a random effects model[27], which takes intra-patient variance caused by repeated Hgb measurements of each patient into account. This indicates that Hgb level measurement error is consistently low across our small, yet diverse study sample size (two subjects suffering from chronic anemia, one healthy male subject, and one health female subject). Additionally, the smartphone Hgb level measurement residual did not correlate with the average between each patient's CBC Hgb level and smartphone Hgb level with ($r = -0.24$), indicating that

residuals were not biased for any specific range of Hgb levels (i.e., algorithm performance remained fairly constant throughout the entire physiologic range of tested Hgb levels) (Fig. 4b). Furthermore, this degree of accuracy falls below a clinically significant threshold for Hgb level measurement of ±1 g dL$^{-1}$ (95% LOA)[22,28,29], suggesting that this system can potentially be considered interchangeable with the CBC Hgb level given further study and an increased sample size[30]. Furthermore, 93% of measurements fall within Clinical Laboratory Improvement Amendment (CLIA) allowable total error of ±7% (difference from CBC)[31], indicating that, upon further refinement and completion of additional testing, this technology may potentially be viable for at-home and clinical use for diagnosis of anemia in addition to the screening capability of the app when uncalibrated.

**Smartphone anemia app is agnostic to sources of interference.** Use of fingernail beds as the imaging area is ideal due to the fact that fingernail beds contain minimal amounts of melanin compared to other parts of the skin[32,33], theoretically rendering this technique insensitive to subject skin tone. To address this experimentally, images were converted into the CIELab color space, a commonly used color quantification system that quantifies color as perceived by the human eye[34]. In particular, the L* value in this color space has been shown to serve as a linear indicator of skin tone[34]. The relationship between the subjects' skin tones and Hgb measurement residuals was determined by measuring the L* value of a patch of skin adjacent to the fingernail. L* did not correlate ($r = 0.13$) with Hgb measurement residual, indicating that subject skin tone has little impact on the ability of the smartphone system to measure Hgb levels (Fig. 5a).

For accessibility in dynamic clinical settings, the smartphone app must function under a wide variety of background lighting conditions. To that end, using luminous flux readings on a digital light meter, no correlation ($r = 0.00$) was found between room brightness and Hgb measurement residual, indicating that this

**Table 2 Diagnosis profile of subjects in clinical assessment**

| Disorder | Number of patients |
|---|---|
| **Hematologic diseases** | 211 |
| *Sickle cell disease* | 162 |
| Hgb SS | 126 |
| Hgb SC | 21 |
| Hgb Sβ+ | 8 |
| Hgb Sβ0 | 5 |
| *Beta thalassemia* | 17 |
| Major | 12 |
| Minor | 5 |
| Thrombocytopenia | 6 |
| Aplastic anemia | 2 |
| Deep vein thrombosis | 2 |
| Hemophilia | 2 |
| Microcytic anemia | 2 |
| Neutropenia | 3 |
| Pulmonary embolism | 2 |
| Anemia (unspecified) | 1 |
| Diamond blackfan anemia | 1 |
| Hemolytic anemia due to immunosuppression | 1 |
| Hyperbilirubinemia | 1 |
| Hypogammaglobulinemia | 1 |
| Iron-deficient anemia | 1 |
| Macrocytic anemia | 1 |
| Normocytic anemia | 1 |
| Pancytopenia | 1 |
| Paroxysmal nocturnal hemoglobinuria (PNH) | 1 |
| Purpura fulminans | 1 |
| Spherocytosis | 1 |
| Von Willebrand disease | 1 |
| **Oncologic diseases** | 54 |
| Acute lymphoblastic leukemia | 29 |
| Hodgkin's lymphoma | 6 |
| Acute myeloid leukemia | 5 |
| Diffuse large B cell lymphoma | 4 |
| Neuroblastoma | 3 |
| Osteosarcoma | 2 |
| Anaplastic ALK-positive large cell lymphoma | 1 |
| Burkitt's lymphoma | 1 |
| Chronic myeloid leukemia | 1 |
| Ewing's sarcoma | 2 |
| Extragonadal germ cell tumor of mediastinum | 1 |
| Germ cell neoplasm of the left testicle | 1 |
| Hepatoblastoma | 1 |
| Lymphoma (unspecified) | 1 |
| Rhabdomyosarcoma | 1 |
| Sacrococcygeal germ cell tumor | 1 |
| Spindle cell sarcoma | 1 |
| Synovial cell sarcoma | 1 |
| T lymphoblastic lymphoma | 1 |
| Wilm's tumor | 1 |
| Healthy Control | 72 |

This study enrolled 337 individuals with a wide variety of diagnoses. Study subjects enrolled consisted of 162 patients with Sickle Cell disease (e.g., type SS, SC, Beta+, Beta 0), 34 patients various other anemias (e.g., Beta Thalassemia major and minor, Microcytic, macrocytic, normocytic, aplastic, iron deficient, and general anemia), 54 instances of several malignancies (e.g., Leukemia, Lymphoma, Sarcoma, Neuroblastoma, Germ Cell Tumor), as well as 15 patients suffering from various other blood conditions (e.g., Deep Vein Thrombosis, hyperbilirubinemia, Hypogammaglobulinemia, idiopathic thrombocytopenic purpura, Purpura fulminans, Pulmonary embolism, Neutropenia, Spherocytosis, Thrombocytopenia, Von Willebrand disease). Additionally, 72 healthy control subjects were enrolled in the study to ensure that a wide range of hemoglobin levels were represented

patch of skin were found to be similar between images taken with smartphones made from different manufacturers and models (Supplementary Figure 6A). Additionally, no statistically significant difference existed between pixel intensity values of fingernail bed images obtained by two different smartphone (Supplementary Figure 6B). Finally, the precision of Hgb level measurements using our technology was found to be ±0.17 g dL$^{-1}$ (± refers to standard deviation of $n = 3$ phones) when tested on multiple images of the same individual's fingernails. Furthermore, preliminary studies suggest that hand temperature and exercise status do not impact Hgb level measurement, indicating that the app may be agnostic to fingernail perfusion variability (Supplementary Table 1).

**App outperforms physician Hgb measurement via physical exam.** Clinical hematologists, US Board certified physicians who specialize in the clinical care of patient with blood disorders, were asked to measure Hgb levels in patients via inspection of images of fingernails. In order to account for physician bias associated with physical examination of patients (e.g., prior knowledge of the patient's medical history), the physicians reviewed the same images of the patients' fingernails as the app. This approach better compares the diagnostic capabilities of physicians and the app. When estimating Hgb levels based on examinations of images of patient fingernail beds ($n = 50$), hematologists estimated blood Hgb levels to within ±4.6 g dL$^{-1}$ of the CBC Hgb level (95% LOA, Fig. 6a). Note that this degree of accuracy represents nearly the entire physiologic Hgb level range tested. The app was then tested on the dataset of 50 patient images and measured Hgb to within ±1.0 g dL$^{-1}$ of CBC Hgb levels (95% LOA, Fig. 6b). Furthermore, ROC analysis revealed an area under the curve of 0.94 for the app vs 0.63 for the hematologists, representing a significant improvement in diagnostic accuracy (Fig. 6c, d). Moreover, agreement of Hgb levels between the physicians' estimates, the smartphone app, and CBC Hgb levels was assessed using the intraclass correlation coefficient (ICC), which found that the smartphone app and CBC Hgb levels demonstrate excellent agreement (defined as ICC < 0.9) as ICC is estimated to be 0.95 (95% confidence interval (CI): 0.92–0.97), while an average of the five hematologists' evaluations demonstrated only moderate agreement with the CBC Hgb levels, with an ICC of 0.59 (0.37–0.74). Importantly, inter-physician variability of Hgb level estimates were high, as indicated by the low level of agreement with an ICC of 0.20 (95% CI .07–0.36).

## Discussion

Given the performance of this technology and high prevalence of anemia worldwide, afflicting nearly two billion people, especially young children, the elderly, and pregnant women, worldwide, this completely noninvasive technology that requires only photos obtained from smartphones has significant implications as a widely accessible screening tool for at risk populations and the general population. The ability to inexpensively diagnose anemia with a high sensitivity, completely noninvasively and without the need for any external smartphone attachments or calibration equipment represents a significant improvement over current POC anemia screening. The external equipment requirements of current existing POC anemia screening technologies represent a significant hurdle for use, as each additional piece of equipment requires a supply chain to support it. For example, even relatively low-cost color calibration cards used to normalize for different background lighting require distribution to the patient and quality control measures regarding the manufacturing process to ensure that the colors are printed precisely and accurately on each card.

In addition, while our system can be used for both anemia screening and diagnosis, it is important to contextualize the

method can be used in a wide variety of settings and lighting conditions (Fig. 5b). Use of the camera flash resulted in the most accurate Hgb level measurement, likely due to the normalization of background lighting conditions provided by the camera flash (Supplementary Figure 5). Furthermore, ensuring that the technology is agnostic to the smartphone make and model, RGB pixel intensity values of the subject's fingernail beds and a control

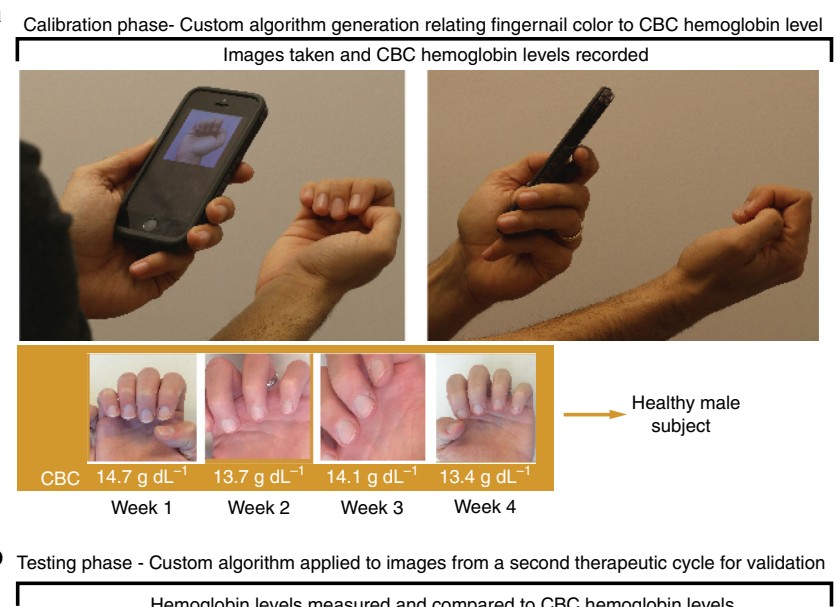

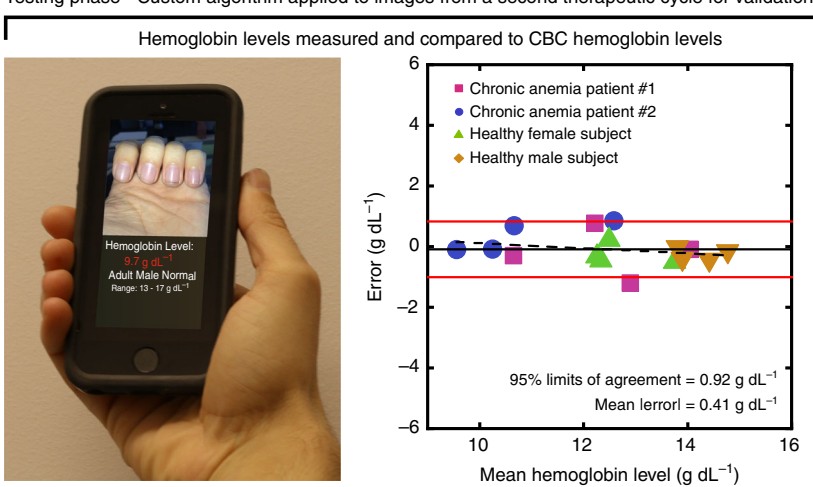

**Fig. 4** Personalized calibration further improves the accuracy of Hgb levels measurement. **a** Healthy and chronically transfused anemic patients were monitored over four weeks (i.e., over the course of a therapeutic blood transfusion cycle). CBC Hgb levels (white text) were used in conjunction with the images to generate a personalized algorithm for each individual. **b** The patient-specific algorithms were used to measure Hgb levels over a subsequent blood transfusion cycle. This patient-specific calibration improved the average error of Hgb level measurements to within 0.41 g dL$^{-1}$ of the CBC Hgb level. Bland–Altman analysis shows negligible experimental bias in the data. A random effects model is used to statistically confirm consistency of average Hgb level measurement error between individual subjects. The average error (solid black line) indicates the Hgb measurement of the smartphone app is negligibly biased. The dashed line represents the correlation ($r = -0.24$) between the residual error and the average of Hgb level measurements obtained from the CBC and the algorithm. The solid red lines represent 95% limits of agreement (0.92 g dL$^{-1}$). $n = 4$ patients, 4 measurements per patient

accuracy requirements of these different clinical scenarios. Though clinical diagnostic tools for anemia have strict accuracy requirements (95% LOA of ± 1.0 g dL$^{-1}$), these requirements are less stringent in POC settings, where anemia screening, rather than diagnosis is crucial. Our results indicate that this smartphone app is ideally suited for screening anemia. Indeed, the accuracy we have presented (95% LOA of ±2.4 g dL$^{-1}$) is comparable or better than currently available POC diagnostic tools such as the invasive Hemocue (95% LOA of ± 2.3 g dL$^{-1}$), the expensive Masimo (95% LOA of ± 3.7 g dL$^{-1}$), and the invasive WHO Color Scale (95% LOA of ± 3.3 g dL$^{-1}$)[19,25,35]. Furthermore, the results of the app when individually calibrated suggest that this technology may, with further study, achieve Hgb measurement accuracy necessary for anemia diagnosis. Going forward, we will continue to increase enrollment in our individual calibration studies to confirm the high level of diagnostic accuracy that would be necessary to truly replace blood-based anemia testing. We present specific use cases highlighting the difference

between anemia screening vs monitoring and diagnosis in Supplementary Figure 7.

Optimizing sensitivity is of paramount importance for a screening tool, due to the ability to correctly identify a high percentage of anemia cases even if this negatively impacts specificity. In its current form, our technology requires the user to simply obtain a fingernail image, which can then be analyzed with an on-board smartphone app that comprises our image analysis algorithm to output the Hgb level measurement or be transmitted remotely to another device (e.g., laptop, desktop computer, cloud-based server with our algorithm embedded into their systems) for remote analysis, the results of which can be immediately transmitted back to the user. After identifying subjects that may possibly be anemic, either type of system can recommend confirmatory Hgb level testing with a CBC, allowing any false positives to avoid unnecessary treatment. Given the ever-increasing rate and near ubiquity of smartphone ownership worldwide[36], this noninvasive, inexpensive, patient-operated Hgb

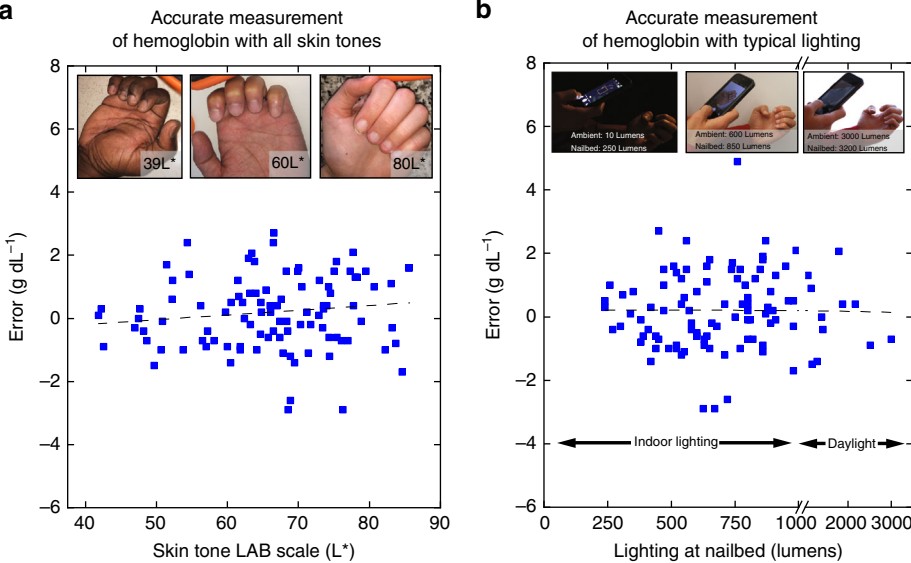

**Fig. 5** Background lighting and subject skin tone has minimal effect on app accuracy. Plotting measurement error against. **a** skin tone and **b** background lighting reveals low and negligible correlation ($r = 0.13$ and $r = 0.00$, respectively) in either case. Dashed lines indicate linear fit between the measurement error and the tested parameter (skin tone and background lighting, respectively). Inset images highlight a representative range of measured background skin tones and lighting conditions. $n = 100$ patients

measurement algorithm allows those at risk of anemia to monitor their conditions using only the native hardware included with their own smartphone[2,3]. This is particularly pertinent in low resource settings, where, contradictory to the relative lack of medical infrastructure, mobile phone networks are quite extensive and have leapfrogged landlines[37].

Additionally, this system has the potential to fundamentally alter the management of patients with chronic anemia. During the course of several weeks, a patient may take images of their fingernail beds and enter their CBC-measured Hgb levels that were obtained as part of their regular outpatient clinical care. Results suggest that these images and Hgb levels may be used to teach the smartphone phone to develop a calibration personalized and tailored to each individual patient. In times of clinical stress, these patients, such as those with genetic causes of anemia or cancer undergoing chemotherapy, may experience rapid, life-threatening, precipitous drops in Hgb and require constant monitoring to determine their need for transfusions. Using this technology, patients could potentially self-monitor their anemia from the comfort of their own home, rather than through inconvenient and recurring clinic visits. In addition, some patients with chronic anemia due to a genetic etiology require chronic transfusions to survive. These scheduled transfusions are currently administered at convenient and regular intervals, and not based on clinical need[38]. Hence, a patient may be transfused too early, exposing them to unnecessary transfusion-related effects (i.e., iron overload, risk of infection), while patients transfused too late may require urgent hospitalization if they develop symptomatic anemia or their Hgb levels decrease to a dangerous level. By enabling continuous and simple monitoring, this technique may empower patients and lead to better allocation of blood bank resources. Moreover, further data collection will increase the size of the patient image pool, facilitating the incorporation of deep machine learning Big Data techniques to further refine the Hgb measurement algorithm[39].

Furthermore, this CBC-validated, smartphone image-based smartphone app for measuring Hgb has the potential to dramatically improve upon the accuracy, cost, and convenience of current Hgb measurement devices while also eliminating the need

for anything other than a smartphone, representing a significant improvement over other POC Hgb measurement technologies[7,8,20,21,40]. With this smartphone image-based Hgb measurement system, any person—healthy or ill—in any location, at any time, now has access to an important health metric and may seek care accordingly. Moreover, healthcare officials in low resource settings may use this technology to inform allocation of limited healthcare resources (e.g., transfusions, high-risk obstetrical services) and medications (e.g., nutritional supplementation such as iron, folate, or vitamin B12)[41] for the patients with the most severe anemia. This completely noninvasive, algorithm-based approach represents a paradigm shift in the way anemia can be screened, diagnosed, and monitored globally. As the system requires no reagents or equipment, the healthcare cost savings could also be significant.

Overall, the ability to conduct self-testing using an unmodified smartphone presents significant advantages over previously reported technologies which require additional equipment such as calibration cards and light-blocking rigs. Moreover, the app utilizes metadata that is automatically obtained from the smartphone camera which enables normalization of background lighting conditions. This presents significant conceptual advantages over existing Hgb measurement technologies, as Hgb levels can now be measured by a patient without requiring a clinic visit or any cumbersome external equipment.

This system suffers from the potential to be impacted by diseases which cause nailbed discolorations such as jaundice and cyanosis[42,43]. However, it is important to point out that a large population of our study subjects suffered from hemolytic anemias, which can lead to jaundice. We found no correlation between disease state and Hgb measurement error, indicating that jaundice is unlikely to impact Hgb measurement (Fig. 4). Furthermore, the image analysis algorithm can potentially be trained in future studies on populations with these disorders to take these discolorations into account. We would also argue that suffering from cardiovascular dysfunction sufficient to cause cyanosis, is a significant enough health problem to render anemia diagnosis a secondary concern, thus obviating the need for these patients to use this app under those circumstances. While these

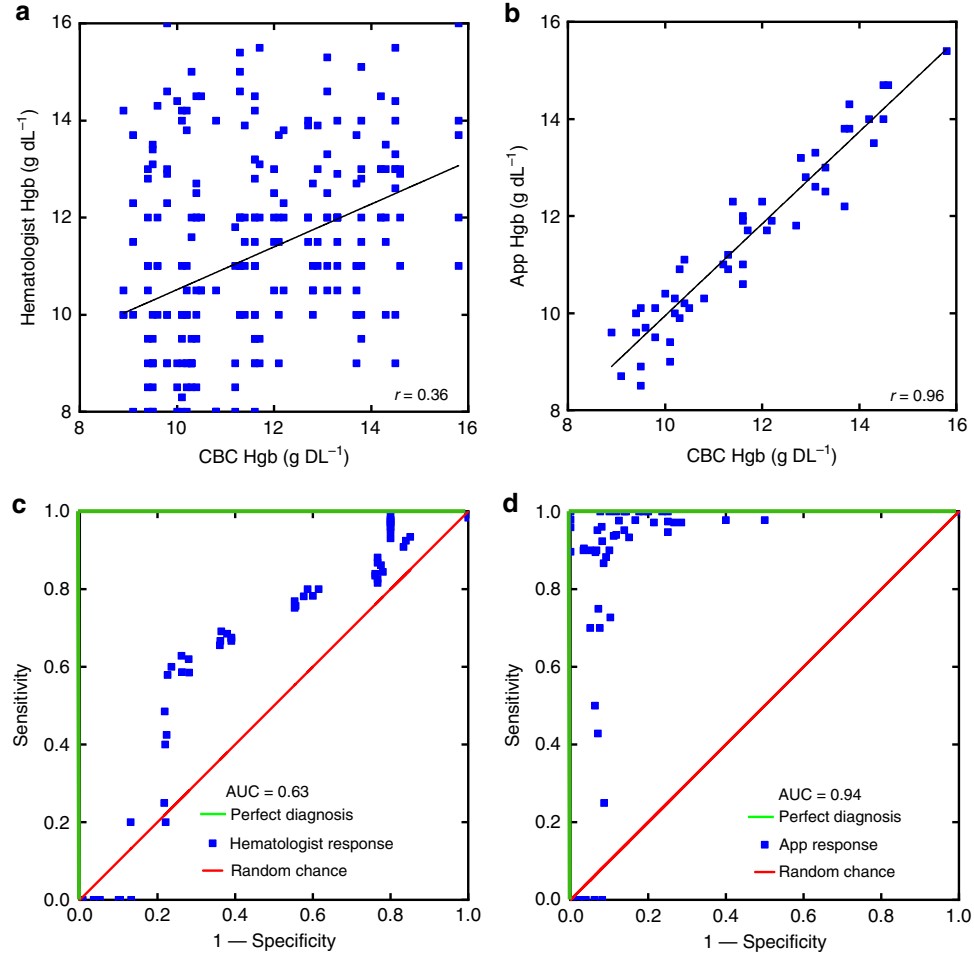

**Fig. 6** App outperforms hematologists in physical exam-based hemaglobin measurement. Hematologists were able to estimate Hgb levels to within ±4.6 g dL$^{-1}$ (**a**) (95% limits of agreement) with an ROC of 0.63 (**c**). The app outperforms the hematologists in both respects with Hgb level accuracy measurement to within ±1.0 g dL$^{-1}$ (**b**) (95% limits of agreement) and an ROC of 0.94 (**d**). $n = 50$ subjects. Plots (**a**, **c**) represent the pooled results of 5 board-certified hematologists estimating blood hemoglobin levels based on images of patients fingernails

conditions may present challenges in Hgb measurement, they present a promising opportunity to use the app to screen for such diseases.

The primary limitations in this study were derived from the use of a single smartphone model and test administrator. Going forward, the potential for user error as well as inter/intra-smartphone variability leading to Hgb measurement error will be addressed in the form of a full clinical assessment, and we will also investigate the efficacy of the smartphone image-based algorithm in which patients will use this app as a self-test using multiple models and manufacturers of smart-phones. This study will also evaluate and improve upon our quality control measures. Overall, the ability to conduct rapid on-demand self-testing demonstrates the versatility of the system and could be especially conducive for global heath applications, where remote diagnosis coupled with tight quality control measures may be preferred and enabled by increasing smartphone use and mobile network prevalence in low resource settings[36]. This approach will shift the anemia screening paradigm worldwide by empowering patients to test themselves from the comfort of their own homes, wherever and whenever they desire.

## Methods
**Clinical assessment of app.** A clinical assessment was conducted at Children's Healthcare of Atlanta, Emory University of School of Medicine, and Georgia Institute of Technology to relate fingernail bed color to Hgb levels. Patients with various anemia etiologies scheduled to have their Hgb levels measured via a CBC as part of their clinical care were recruited to this study ($n = 265$). Subjects were excluded by quality control measures if their images showed fingernail beds that were obscured or discolored due to leukonychia, nailbed injury, nail polish, darkening due to medication[44], etc. Exclusions were conducted to eliminate unnecessary variables that could obfuscate algorithm development. All CBC's were conducted using blood samples collected via venous blood draw. After patient's blood was collected to conduct their CBCs, two images were taken of those patients' fingernail beds. Smartphone pictures were obtained with the camera flash both on and off. All images were taken with an Apple iPhone 5s (Apple, Cupertino, CA) using all default imaging settings. Prior to imaging, the auto-focus and brightness adjustment of the smartphone camera was activated by tapping the screen in order to focus on the nailbed. To ensure consistent images, each image was taken with the smartphone at a distance of ~0.5 m from the subjects' fingernail beds. If possible, subjects were encouraged to curl their fingers inwards with their palms facing upwards to control for possible alterations in blood flow caused by hand and finger positioning that could potentially affect the underlying color of the fingernail beds (Fig. 1). Images were taken in clinic examination rooms, where lighting conditions and room illuminants were relatively consistent. A digital light meter (Hisgadget, Union City, CA) was placed next to the subject's fingernail beds to further ensure consistent background lighting conditions, but external digital light meter readings were not incorporated into the Hgb level calculation. An additional 72 healthy subjects from Emory University and The Georgia Institute of Technology were tested using an identical protocol. CBC's were conducted on each subject prior to imaging and were analyzed via the same clinical hematology analyzer (Advia 2120i, Siemens, Berlin, Germany) used in the clinical study. All imaging was conducted in a room with similar lighting conditions to the clinic exam rooms, which was confirmed via digital light meter. Fingernail bed images and blood Hgb levels were analyzed in a total of 337 subjects. These subjects' blood Hgb levels ranged between 5.9 and 16.8 g dL$^{-1}$ (Supplementary Figure 8A). Subject's ages ranged between 1 and 60 years old and had varying skin tones (Supplementary Figure 8B). 167 female subjects and 170 male subjects were enrolled in this study. In six cases, fingernail polish was discovered after informed consent had

been obtained, and these subjects were excluded from testing after study enrollment. In one case, an image labeled as having been taken with the camera flash on was discovered to have been taken with the flash off, resulting in this subject's data being excluded.

**Algorithm development/image processing**. Smartphone images were transferred or transmitted from the smartphone used in the study to a computer. Fingernail data, skin color data, and image metadata were extracted from fingernail bed smartphone images via MATLAB (Mathworks, Natick, MA). Regions of interest, from which fingernail and skin color data were extracted, were manually selected to ensure that fingernail irregularities were excluded from analysis. These regions of interest were selected from each finger, excluding the thumb, and were 900 pixels$^2$, corresponding to ~10 mm$^2$ on the fingernail. Color data were extracted from each region and averaged together across fingers for each subject. This was shown to be an acceptable method due to the low color variability between different fingers (Supplementary Figure 9). An algorithm was then written in MATLAB utilizing robust multi-linear regression with a bisquare weighting algorithm to relate the image parameter data to CBC Hgb levels for each patient (Eq. (1))[45].

Hemoglobin$_{Result} = C + P_1 \times W_1 + P_2 \times W_{2+} \ldots P_n \times W_n$ where: $C$ = constant, $W$ = weights determined via robust multi linear regression, and $P$ = skin color data, and image metadata parameters.

A uniform bias adjustment factor was also added to address the inherent variability in fingernail measurement. Two distinct use models and algorithms were applied for this Hgb measurement method: (1) as a noninvasive, smartphone-based, quantitative Hgb level diagnostic requiring calibration with CBC Hgb levels that enables chronic anemia patients to self-monitor their Hgb levels, and (2) as a noninvasive, smartphone-based anemia screening test that does not require calibration with CBC Hgb levels. Sampling strategies were used to generate the algorithm depending on the specific application.

Anemia screening among the general population: To develop the algorithm as a tool to screen for anemia, the entire study population (337 subjects) was randomly split into a discovery group (237 subjects) and a testing group (100 subjects). The discovery group was used to establish the relationship between image parameters and Hgb levels via robust multi-linear regression, much like the calibration phase of the personalized calibration study. A testing group, analogous to the testing phase of the personalized calibration study, of 100 subjects was used to validate the resultant algorithm. Validation was performed by applying the smartphone algorithm to each testing image and comparing the algorithm generated Hgb result with the CBC Hgb result (i.e., determining the residual of the algorithm-based method). This process was repeated 1000 times with different, randomly-selected without replacement, discovery/testing groups to minimize residual error, thereby optimizing the parameters of the algorithm for anemia screening. Resulting data from most accurate outcome of this optimized screening algorithm is depicted in Fig. 2. Hgb measurements taken from the previously described personalized calibration study were not included in this anemia screening study.

Personalized calibration of Smartphone App: A personalized calibration approach was tested in two β-thalassemia major patients with chronic anemia currently undergoing chronic transfusion therapy, a healthy female subject with Hgb levels that fluctuated during her menstrual cycle, and a healthy male subject with consistent Hgb levels over an identical timeframe to assess the algorithm's capability to be accurately personalized and calibrated to that individual, regardless of their diagnosis or Hgb levels. Treatment for β-thalassemia major currently comprises red blood cell transfusions to compensate for the patients' ineffective erythropoiesis[46]. Hgb levels in the chronic anemia patients fall throughout a 4-week transfusion cycle, which was chosen as an appropriate time interval for this study. Smartphone Images were obtained with and without the camera flash. Prior to each imaging session, CBC Hgb levels were obtained from each subject via venipuncture. Color data and phone metadata were compiled and a relationship between image data and CBC Hgb levels was established via robust multi-linear regression. This process was repeated for each individual using data from the 4 weeks of images to create a unique calibration curve personalized for that individual. Image parameter changes associated with Hgb level fluctuations specific to each person were related to perform algorithm calibration specific to each subject, thus improving the accuracy of Hgb level estimation. After the smartphone image analysis system was calibrated for each subject, Hgb levels were measured weekly over the next 4 weeks using the newly personalized algorithm. These Hgb level measurements were then compared to the CBC Hgb levels obtained at the same time to assess accuracy. This personalized calibration occurred over a total of 8 weeks.

**Hemoglobin measurement from images of fingernails**. Images were taken of 50 subjects fingernails from the previously described clinical study. These subjects' ages ranged from 1 to 62 years old. Hematologists (M.D. physicians who specialize in clinical hematology and are trained and Board Certified in the USA) were instructed to analyze each image and measure Hgb levels. For comparison, images were loaded into the app, and the Hgb measurement protocol was performed on these images. All images and analysis were taken using an iPhone 5S. It is important to note that these images were not used in the development of the underlying image analysis algorithm.

Intraclass correlation coefficient (ICC) reflects not only degree of correlation but also agreement between measurements and ranges between 0 and 1, with values closer to 1 representing stronger reliability. Reliability refers to the degree of agreement among raters. It gives a score of how much homogeneity, or consensus, there is in the ratings given by different judges or instruments. The ICC is able to incorporate the reliability of more than 2 raters-as in the case of the 5 hematologists evaluating nail beds. Patients and the physicians were assumed to be random samples from the respective populations they represent.

**App development**. The Hgb level measurement algorithm was incorporated into mobile apps. The open source integrated development environment (IDE) Android Studio (Google, Mountain View, CA) was used to develop a beta version of the Hgb measurement app in the Android operating system. The proprietary IDE Xcode (Apple, Cupertino, CA) was used to develop a beta version of the app in the iOS operating system.

**Human subjects research statement**. All experiments complied with all relevant ethical guidelines for human subject research, namely, the Declaration of Helsinki. Verbal assent and written consent were obtained from all study subjects and their parents (age permitting) in accordance with HIPAA regulations prior to partaking in the study. All experiments involving human subjects in this manuscript were approved by either the Emory University IRB (algorithm development—approval number 00081226) or the Georgia Institute of Technology IRB (skin temperature and heart rate interference—approval number H17118).

**Statistical analysis**. Statistical significance ($p < 0.05$) was determined via two-tailed Student's $t$-test assuming unequal variance. All statistical tests (calculation of regression correlation coefficients and Student's $t$-tests were conducted using Origin Pro 2017 student version. 95% confidence intervals for sensitivity and specificity are calculated according to the efficient-score method. A two-way random effects model was used to estimate our ICC for measuring agreement between the app and hematologists at measuring Hgb levels based on physical examination.

**Code availability**. The custom MATLAB code used in this study is available from the corresponding author upon reasonable request. This code is copyrighted by Emory University and Children's Healthcare of Atlanta and is to be used only for educational and research purposes. Any commercial use including the distribution, sale, lease, license, or other transfer of the code to a third party, is prohibited. For inquiries regarding commercial use of the code, please contact Emory University's Office of Technology Transfer.

## Data availability

The de-identified datasets collected and analyzed in this study (i.e., images of fingernail beds and the surrounding skin) and any remaining data are available from the corresponding author on reasonable request.

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

## Acknowledgements

Financial support for this work was provided by a National Science Foundation Graduate Research Fellowship DGE-1650044 (to R.G.M.), a National Institutes of Health R21 (EB025646-01), and the 2017 Massachusetts General Hospital Primary Care Competition Prize. Also, this work was performed in part at the Georgia Tech Institute for Electronics and Nanotechnology, a member of the National Nanotechnology Coordinated Infra-structure, which is supported by NSF ECCS (1542174).

## Author contributions

All authors collaborated to design the experiments and clinical study. R.G.M., E.A.T., W. A.L., and J.B. facilitated subject enrollment. R.G.M. performed experiments, conducted the clinical study, and analyzed the data. R.G.M. and T.L. conducted statistical analysis. R.G.M., D.R.M., E.A.T., C.C., G.C., and W.A.L. wrote the paper. All authors contributed to editing the paper.

## Additional information

**Competing interests:** The authors declare no competing interests.

