## [Peer Review File · Nature Communications]

Reviewers' comments:

Reviewer #1 (Remarks to the Author):

The Authors report on a non-invasive diagnostics that may replace common blood-based laboratory tests requiring only patient-sourced smartphone photos, detecting anemia (hemoglobin levels <12.5 g/dL) with a sensitivity of 94% (95% CI, 89% – 100%) when compared with CBC hemoglobin levels (N=100 subjects)

Implemented as either a smartphone app or a cloud-based image analysis algorithm, this technology is broadly adaptable for other blood-based biomarkers, has significant potential to reduce healthcare costs, and is conducive for global health applications

Comment: very interesting approach to the diagnosis of anemia in large patient populations, especially with limited access to health care resources.

The CI of 2.5 gr of H bis not totally irrelevant (5-7-9 gr Hb makes quite a difference

One would like to know how much better this smrtphone app is, as compared to visual judgement of an experienced GP or hematologist: it would ony require 20 subject evaluated with both methods and compare the results

Reviewer #2 (Remarks to the Author):

The manuscript describes an alternative method for measuring blood hemoglobin concentrations through nailbed hue colorimetric analysis. Patient-sourced images taken with a smartphone are analyzed on the smartphone's native hardware. The technique is shown to be robust across variable conditions of lighting, skin tone, skin temperature and smartphone models with additional improvements to performance through personal calibration of individual patients. This work leverages a novel diagnostic "marker" of nailbed hue and the smartphone as a platform device to illustrate a technique that could be well suited for point-of-care deployment. If the results are valid, this would be a very novel and exciting new method for anemia diagnosis.

Statistical analysis and description of performance could be clarified with clearer descriptions of sample size used in each set of calculations. Accuracy (in the personal calibration case) is reported as ± 0.90 g/dL of CBC hemoglobin levels, using a sample size of $N=4$ individuals. This sample size seems insufficient for convincing statistical analysis of accuracy. The authors mention that clinically significant threshold is ± 1 g/dL, however in the non-calibrated general use study, Hgb levels were reported to be measured to within ± 2.6 g/dL of CBC hemoglobin levels using a sample size of $N=100$ patients. Thus for the broader case and larger representative sample size, are the limits of agreement range adequate, given the clinically significant threshold? Finally, the authors should report the Figure 2 performance values in the abstract instead of the personal calibration use-case values, as they are more representative of device/method performance.

The manuscript could be strengthened with additional discussion on the clinical merit of nailbed hue as a method for measuring hemoglobin concentration. Are there confounding variables or other medical conditions that could affect nailbed hue (and thus result in false positives or negatives)?

Minor comment:

-Supplementary Figure S4: number of anemia positive individuals should be 66 and not 77 in the Final diagnosis box

Reviewer #3 (Remarks to the Author):

The manuscript by Mannino et al describes an imaging-based approach to measure blood hemoglobin (Hgb) levels. Specifically, the authors used a smartphone to take photos of fingernail beds, and developed customized image processing algorithms to estimate Hgb levels. The entire assay can be performed using the smartphone only, not requiring additional attachments or calibration tools. Results obtained by assessing the Hgb levels from the fingernail beds were quantitatively comparable to those obtained with a conventional complete blood count (CBC). Additional personalized calibration based on the CBC as a reference increases accuracy of the Hgb level analysis. While the current work presents a good level of technical advances, its major concepts are hardly new:

1. Using physical examination (color tint of the lower eyelid conjunctiva, nail-bed rubor, nail-bed blanching, and palmar crease rubor) to diagnose anemia has been known as shown in Ref. 17

2. Using a smartphone for point-of-care anemia diagnosis has been reported (Ref. 36 PLoS One 11:e0153286), which analyzed photos of conjunctival pallor. The achieved accuracy (AUC, 0.86) was similar to the current work (AUC, 0.87).

3. There are several FDA-approved POC devices (Masimo Pronto-7, OrSense NBM200) for non-invasive blood analyses. These devices not only check Hgb levels but also other indicators such SpO₂, perfusion, pulse rate.

In the line of these assessments, the current work might be better suited for technical journals.

Other minor comments:

- Please provide a reference to the Hgb level cutoff (12.5 g/dL) used in the manuscript. Official WHO criteria indicate different populations have different Hgb cutoff levels on anemia. The data set should be reorganized or additional data set is required to claim the capability of the system for the diagnosis of anemia considering the population and its corresponding Hgb level cutoff.

- What is the processing time of the photos to get the Hgb level?

We thank the reviewers for the careful consideration of our manuscript and the suggested experiments and clarifications that further strengthen our work. We are extremely encouraged that the referees stated that our technology has “*significant potential to reduce healthcare costs, and is conducive for global health applications*” and presents “*a good level of technical advances*”. However, the reviewers’ enthusiasm for our approach was somewhat dampened by a lack of clarity on: 1) the novelty of this approach, 2) how our technology compares to existing published work, and 3) the actual development of a smartphone app. Accordingly, we have addressed all of these concerns in an updated manuscript that includes substantial improvements, specifically we have:

1. Updated Figure 1, added Supplemental Video 1, and added manuscript text that better showcases the fully developed and functional smartphone anemia app. Specific ambiguous language such as “smartphone-based image-analysis algorithm” was replaced with “smartphone app” to avoid confusion over the creation of an actual app.
2. Added a new Figure 6 to demonstrate that the smartphone anemia app significantly outperforms board-certified physician hematologists in performing assessment of anemia via physical examination with 95% limits of agreement (LOA) of ± 1.0 g/dL for our smartphone anemia app versus ± 4.6 g/dL of the physicians’ physical exam anemia assessments compared to gold standard hemoglobin (Hgb) levels obtained via complete blood count (CBC)
3. Added a new Table 1 to better demonstrates the unique advantages of our smartphone anemia app technology compared to all existing Hgb level assessment tests that have been published or are currently on the market
4. Included new manuscript updates that better highlight the novelty of this system, namely, that ours is the first system to leverage smartphone photo metadata in our completely non-invasive measurement of Hgb levels, and that our technology requires ONLY the smartphone itself and eliminates the need for any external equipment, including smartphone attachments or calibration cards, which always present a significant logistical burden for “on demand” use or use in low resource settings.
5. Added a new Figure 7 that better highlights how eliminating the need for any external equipment aside from a smartphone enables novel usage models that are uniquely suited for anemia screening in low resource as well as developed areas of the world and for chronically anemic patient populations to serially and non-invasively monitor their disease as frequently as they desire.

All changes to the original manuscript are highlighted in yellow in the revised manuscript and in the responses below. All reference numbers cited by the editor, reviewers, and our responses have been changed to reflect their position in the current manuscript, as the reference order has changed from the initial submission. Please see the response below for our point-by-point responses to the reviewer’s comments.

Reviewer #1 (Remarks to the Author):

The Authors report on a non-invasive diagnostics that may replace common blood-based laboratory tests requiring only patient-sourced smartphone photos, detecting anemia (hemoglobin levels <12.5 g/dL) with a sensitivity of 94% (95% CI, 89% – 100%) when compared with CBC hemoglobin levels (N=100 subjects) Implemented as either a smartphone app or a cloud-based image analysis algorithm, this technology is broadly adaptable for other

blood-based biomarkers, has significant potential to reduce healthcare costs, and is conducive for global health applications.

Comment: very interesting approach to the diagnosis of anemia in large patient populations, especially with limited access to health care resources. The CI of 2.5 gr of Hb is not totally irrelevant (5-7-9 gr Hb makes quite a difference)

We would like to thank the reviewer for this astute observation. The reviewer raises an excellent point, namely, that it is important to contextualize the accuracy in terms of the clinical use case of the application. Anemia detection can be broadly categorized into 2 primary clinical use cases, anemia diagnosis and anemia screening. Each of these use cases has different accuracy requirements. In clinical settings, where highly accurate CBC measurement of Hgb levels is available to accurately diagnose anemia, a CI of 2.5 g/dL can, indeed, be significant, as the reviewer noted. However, in POC settings such as home-based or global health settings, where access to clinical-grade tests are obviously limited (as the reviewer astutely points out), a CI of 2.5 for an anemia screening test could be tolerated and users who are screened as anemic and then referred for further and more definitive confirmatory testing with CBC's in a clinical or hospital setting. This concept is validated in studies such as the study in ref 19 (Moore *et al. American Journal of Surgery* 2013). Thus, we believe that our system is a useful tool in POC or home-based settings for anemia screening as we have achieved accuracy levels on par with, or greater than, currently available technologies currently used for POC anemia screening. We have strengthened this argument in the manuscript with the following additional sentences in the manuscript:

Main Text: Anemia has numerous causes, ranging from common nutritional causes, such as iron or folate deficiency, which are relatively straightforward to treat and cure, to rarer genetic causes, such as sickle cell disease or thalassemia major, which lead to severe and chronic anemia that requires frequent monitoring. Detection of anemia involves either anemia screening or anemia diagnosis, and both require different degrees of measurement accuracy. First, a clear clinical need exists for easily and widely accessible tools to screen for anemia among at-risk populations (e.g., pregnant women, toddler-age children, elderly patients) or the general public to determine whether an individual will need formal confirmatory testing with the gold standard Hgb level test obtained via a complete blood count (CBC). However, there is also a need to for non-invasive methods to more quantitatively and officially diagnose and monitor anemia with higher precision Hgb levels, especially patients with known or chronic anemia. Here we describe an approach that has the potential to significantly improve both the screening and diagnosis of anemia.

Results: In fact, this degree of accuracy is on par with reported accuracy values in POC settings of the invasive clinically used Hemocue and substantially better than under development POC screening tools such as HemaApp and conjunctival analysis via photographs.

Discussion: Though clinical diagnostic tools for anemia have strict accuracy requirements (± 1.0 g/dL), these requirements are less stringent in POC settings, where anemia screening, rather than diagnosis is crucial. Our results indicate that this smartphone app is ideally suited for screening anemia. Indeed, the accuracy we have presented (± 2.4 g/dL) is comparable or better than currently available POC diagnostic tools such as the invasive Hemocue (± 2.3 g/dL), the expensive Masimo (± 3.7 g/dL), and the invasive WHO Color Scale (± 3.3 g/dL). Furthermore, the results of the app when individually calibrated suggest that this technology may, with further

study, achieve Hgb measurement accuracy necessary for anemia diagnosis. We present specific use cases highlighting the difference between anemia screening vs monitoring in Fig. 7.

One would like to know how much better this smrtphone app is, as compared to visual judgement of an experienced GP or hematologist: it would ony require 20 subject evaluated with both methods and compare the results

We would like to thank the reviewer for this insightful experimental suggestion that enables us to properly contextualize the accuracy of our system. Our new experimental data show that the app can significantly outperform the diagnostic capabilities of 5 trained hematologists (M.D. physicians who specialize in clinical hematology and are trained and Board Certified in the U.S.A.) when examining images of patient's fingernails ($n = 50$). When estimating Hgb levels based on physical examinations of fingernail beds, trained hematologists were accurate to within ± 4.6 g/dL of the gold standard CBC Hgb level, which represents nearly the entire Hgb level range tested. On the same subset of 50 patients, the app was able to measure Hgb to within ± 1.0 g/dL of CBC Hgb levels, representing a significant improvement. Furthermore, ROC reveals an area under the curve of 0.94 for the app vs 0.63 for the hematologists, representing a significant improvement in diagnostic accuracy. Moreover, we analyzed agreement of Hgb levels between the physicians, app, and CBC using the intraclass correlation coefficient (ICC) and found that the App and CBC demonstrate excellent agreement (defined as $ICC < .9$) as ICC is estimated to be 0.95 (95% confidence interval (CI): 0.92 - 0.97), while an average of the 5 hematologists evaluations demonstrate only moderate agreement with the CBC with an ICC of 0.59 (0.37 - 0.74). It is important to note that the physicians results do not agree with each other, as indicated by the level of agreement between each hematologist's results is poor with a ICC of 0.20 (95% CI .07 - 0.36). The following new paragraphs and figure have been added to the methods, results, and discussion section of the revised manuscript describing this experiment and the results and implications thereof:

Results: The app outperforms clinical hematologists' ability to measure Hgb levels via physical examination.

Clinical hematologists, US Board certified physicians who specialize in the clinical care of patient with blood disorders, were asked to measure Hgb levels in patients via inspection of images of fingernails. In order to account for physician bias associated with physical examination of patients (e.g. prior knowledge of the patient's medical history), the physicians reviewed the same images of the patients' fingernails as the app. This approach better compares the diagnostic capabilities of physicians and the app. When estimating Hgb levels based on examinations of images of patient fingernail beds ($n = 50$), hematologists estimated blood Hgb levels to within ± 4.6 g/dL of the CBC Hgb level (Fig 6A). Note that this degree of accuracy represents nearly the entire physiologic Hgb level range tested. The app was then tested on the dataset of 50 patient images and measured Hgb to within ± 1.0 g/dL of CBC Hgb levels (Fig 6B). Furthermore, ROC analysis revealed an area under the curve of 0.94 for the app vs 0.63 for the hematologists, representing a significant improvement in diagnostic accuracy (Fig 6C-D). Moreover, agreement of Hgb levels between the physicians' estimates, the smartphone app, and CBC Hgb levels was assessed using the intraclass correlation coefficient (ICC), which found that the smartphone all and CBC Hgb levels demonstrate excellent agreement (defined as $ICC < .9$) as ICC is estimated to be 0.95 (95% confidence interval (CI): 0.92 - 0.97), while an average of the 5 hematologists' evaluations demonstrated only moderate agreement with the CBC Hgb levels, with an ICC of 0.59 (0.37 - 0.74). Importantly, inter-physician variability of Hgb level

estimates were high, as indicated by the low level of agreement with an ICC of 0.20 (95% CI .07 - 0.36).

Fig. 6: Smartphone app outperforms trained hematologists in hemaglobin measurement based on physical examination. Hematologists were able to estimate Hgb levels to within ± 4.6 g/dL (A) (95% limits of agreement) with an ROC of 0.63 (C). The app outperforms the hematologists in both respects with Hgb level accuracy measurement to within ± 1.0 g/dL (B) (95% limits of agreement) and an ROC of 0.94 (D). $n = 50$ subjects. Plots (A,C) represent the pooled results of 5 board certified hematologists estimating blood hemoglobin levels based on images of patients fingernails.

Methods: Hemoglobin measurement from images of fingernails

Images were taken of 50 subjects fingernails from the previously described clinical study. These subjects' ages ranged from 1 to 62 years old. Hematologists (M.D. physicians who specialize in clinical hematology and are trained and Board Certified in the U.S.A.) were instructed to analyze each image and measure Hgb levels. For comparison, images were loaded into the app, and the Hgb measurement protocol was performed on these images. All images and analysis were taken using an iPhone 5S. It is important to note that these images were not used in the development of the underlying image analysis algorithm.

Intraclass correlation coefficient (ICC) reflects not only degree of correlation but also agreement between measurements and ranges between 0 and 1, with values closer to 1 representing stronger reliability. Reliability refers to the degree of agreement among raters. It gives a score of how much homogeneity, or consensus, there is in the ratings given by different judges or instruments. The ICC is able to incorporate the reliability of more than 2 raters-as in the case of the 5 hematologists evaluating nail beds. Patients and the physicians were assumed to be random samples from the respective populations they represent.

Reviewer #2 (Remarks to the Author):

The manuscript describes an alternative method for measuring blood hemoglobin concentrations through nailbed hue colorimetric analysis. Patient-sourced images taken with a smartphone are analyzed on the smartphone's native hardware. The technique is shown to be robust across variable conditions of lighting, skin tone, skin temperature and smartphone models with additional improvements to performance through personal calibration of individual patients. This work leverages a novel diagnostic "marker" of nailbed hue and the smartphone as a platform device to illustrate a technique that could be well suited for point-of-care deployment. If the results are valid, this would be a very novel and exciting new method for anemia diagnosis.

Statistical analysis and description of performance could be clarified with clearer descriptions of sample size used in each set of calculations. Accuracy (in the personal calibration case) is reported as +/- 0.90g/dL of CBC hemoglobin levels, using a sample size of N= 4 individuals. This sample size seems insufficient for convincing statistical analysis of accuracy. The authors mention that clinically significant threshold is +/- 1 g/dL, however in the non-calibrated general use study, Hgb levels were reported to be measured to within +/- 2.6g/dL of CBC hemoglobin levels using a sample size of N= 100 patients. Thus for the broader case and larger representative sample size, are the limits of agreement range adequate, given the clinically significant threshold? Finally, the authors should report the Figure 2 performance values in the abstract instead of the personal calibration use-case values, as they are more representative of device/method performance.

We would like to thank the reviewer for raising an important point regarding characterization of Hgb diagnostic accuracy. In the original manuscript, we failed to adequately put the accuracy of our app in the appropriate clinical context. It is important to contextualize the error based on the 2 major use cases for anemia detection, namely, screening vs. diagnosis. Each of these use cases requires a different degree of accuracy. We mention the 1 g/dL accuracy figure as that is the clinically acceptable standard for a hospital-based diagnosis and the results of the personalized study fall within these limits, suggesting potential of our app reaching these levels if properly calibrated. We agree with the reviewer's suggestion that we should report the Figure 2 performance values in the abstract and have made this change in the manuscript. Furthermore, we implemented new quality control algorithms into the app which improved accuracy from ± 2.6 g/dL to ± 2.4 g/dL. We make the argument that this level of accuracy is acceptable for anemia screening in POC settings by comparing this level of accuracy to POC devices such as the hemocue (± 2.3 g/dL, ref 25 Jaggernath *et al. PLOS ONE* 2016), Masimo Pronto 7 (± 3.7 g/dL, ref 19 Moore *et al. American Journal of Surgery* 2013) and the World Health Organization (WHO) color scale (± 3.3 g/dL, ref 34 Paddle Bulletin of the WHO 2002) all of which require additional equipment, are costly, or are invasive and require blood sampling. Furthermore, we have addressed qualitative comparisons in a new Table 1. Finally, we have included an additional figure 7 that describes the proposed use cases of anemia screening vs diagnosis. Additional

references were added to support these accuracy comparisons (References 19, 25, and 34). The additional comments, table and figure in the manuscript can be found below:

Device	Non-invasive	Smartphone-Based	No Additional Equipment (aside from smartphone)	Low-Cost	Accurate
Complete Blood Count (Gold Standard)	X	X	X	X	✓
Hemocue	X	X	X	X	✓
Masimo Co-Oximetry	✓	X	X	X	X
Conjunctival Analysis	✓	✓	X	✓	X
WHO Color Scale	X	X	X	✓	X
HemaApp	✓	✓	X	✓	X
Smartphone Anemia App	✓	✓	✓	✓	✓

Table 1: Comparison table of currently used anemia diagnostic technologies with the smartphone app

A Anemia screening in low resource setting

B Screening at risk patients in developed nations

C Monitoring of chronically anemic patients

Fig. 7: The smartphone app enables anemia screening and patient self-monitoring of Hgb levels in different use cases. Our smartphone app can be used for both anemia screening (A, B) and anemia self-monitoring (C) in both economically-developed and low resource settings. **A)** In low resource settings, a healthcare worker can download the app and take their smartphone into the field for anemia screening of specific population or community. Results are then used to assist healthcare workers with allocation of scarce medical resources to patients who need them most. **B)** In the setting of a developed nation, a pregnant woman, for example, who suddenly experiences fatigue and suspects she may be anemic can download the app at home and screen herself for anemia in under a minute. The “on demand” anemia self-screening capability of this app will then allow her to transmit her results to her physician for guidance as to whether she requires further confirmatory testing and treatment at her local hospital or clinic. **C)** In the case of the chronically anemic patient, the physician prescribes the app for Hgb level tracking over time. CBC Hgb levels will first be obtained from the patient over several clinic visits to calibrate the Hgb measurement algorithm. Once calibrated and “personalized” for each patient, the smartphone app can be used to obtain frequent, non-invasive Hgb levels measurements whenever the patient desires. These Hgb level results, and their trends thereof, can then be transmitted to the physician to adjust medication dosages or to better personalize the timing of treatments such as blood transfusions.

Main Text: Anemia has numerous causes, ranging from common nutritional causes, such as iron or folate deficiency, which are relatively straightforward to treat and cure, to rarer genetic causes, such as sickle cell disease or thalassemia major, which lead to severe and chronic anemia that requires frequent monitoring. Detection of anemia involves either anemia screening or anemia diagnosis, and both require different degrees of measurement accuracy. First, a clear clinical need exists for easily and widely accessible tools to screen for anemia among at-risk populations (e.g., pregnant women, toddler-age children, elderly patients) or the general public to determine whether an individual will need formal confirmatory testing with the gold standard Hgb level test obtained via a complete blood count (CBC). However, there is also a need to for non-invasive methods to more quantitatively and officially diagnose and monitor anemia with higher precision Hgb levels, especially patients with known or chronic anemia. Here we describe an approach that has the potential to significantly improve both the screening and diagnosis of anemia.

Results: In fact, this degree of accuracy is on par with reported accuracy values in POC settings of the invasive clinically used Hemocue and substantially better than under development POC screening tools such as HemaApp and conjunctival analysis via photographs.

Discussion: Though clinical diagnostic tools for anemia have strict accuracy requirements (± 1.0 g/dL), these requirements are less stringent in POC settings, where anemia screening, rather than diagnosis is crucial. Our results indicate that this smartphone app is ideally suited for screening anemia. Indeed, the accuracy we have presented (± 2.4 g/dL) is comparable or better than currently available POC diagnostic tools such as the invasive Hemocue (± 2.3 g/dL), the expensive Masimo (± 3.7 g/dL), and the invasive WHO Color Scale (± 3.3 g/dL). Furthermore, the results of the app when individually calibrated suggest that this technology may, with further study, achieve Hgb measurement accuracy necessary for anemia diagnosis. We present specific use cases highlighting the difference between anemia screening vs monitoring in Fig. 7.

The manuscript could be strengthened with additional discussion on the clinical merit of nailbed hue as a method for measuring hemoglobin concentration. Are there confounding variables or other medical conditions that could affect nailbed hue (and thus result in false positives or negatives)?

We would like to thank the reviewer for this excellent point and for the opportunity to better describe the clinical merit of the nailbed as well as the opportunity to include a thorough discussion of other medical conditions that may impact nailbed hues.

The nailbeds are an ideal body region to image for anemia measurement for a number of reasons. From a purely usability and photography-based perspective, the nailbeds offer a very large and consistent imaging window that is similar amongst individuals. From a usability perspective, it is very easy for an individual to take a photo of their own fingernails using their smartphone. This is a key advantage as the ability to conduct self-testing is one of our key innovations. Finally, the nailbeds do not contain melanin pigment producing cells, allowing for skin tone-agnostic Hgb measurement. We have included several statements in the manuscript highlighting these facts.

However, as the reviewer pointed out, several medical conditions are known to affect nailbed hue, which may potentially impact Hgb level measurement using our app. The most prevalent conditions known to cause changes to the nailbed hue include: cyanosis, jaundice, and hemolytic disorders. Cyanosis results in blue discoloration of the nailbeds and can be caused by a variety of conditions that lead to poor circulation (e.g. pulmonary diseases, heart failure) potentially

impacting Hgb results if a patient is tested with our app. However, anemia detection is a secondary priority for patients suffering from these serious cardiovascular events that lead to cyanosis. Jaundice, on the other hand, manifests in yellowing of the fingernail beds and can be caused by disorders such as liver disease or hemolytic disorders causing excessive red blood cell (RBC) destruction. While these fingernail discolorations are possible, we have actually tested a large group of individuals with hemolytic anemia, and have shown that the diagnosis of the individual does not impact Hgb level measurement. Future studies will be conducted to investigate the impact of these disorders on Hgb measurement. Furthermore, the underlying method of converting color data into disorder detection employed by this app may potentially be used to diagnose these disorders as well. This is all described in a new paragraph in the discussion section.

The following additions have been made to the manuscript:

Main Text: Since fingernails, conjunctiva, and palmar creases do not contain melanocytes (melanin producing skin cells), the primary source of color of these anatomical features is blood Hgb¹⁸. Of these sites, fingernails are straightforward for a user to self-image, unlike conjunctiva, and also have low person-to-person size and shape variability, unlike palmar creases.

Discussion: This system suffers from the potential to be impacted by diseases which cause nailbed discolorations such as jaundice and cyanosis. However, it is important to point out that a large population of our study subjects suffered from hemolytic anemias, which can lead to jaundice. We found no correlation between disease state and Hgb measurement error, indicating that jaundice is unlikely to impact Hgb measurement (Fig. 3). Furthermore, the image analysis algorithm can potentially be trained in future studies on populations with these disorders to take these discolorations into account. We would also argue that suffering from cardiovascular dysfunction sufficient to cause cyanosis, is a significant enough health problem to render anemia diagnosis a secondary concern, thus obviating the need for these patients to use this app under those circumstances. While these conditions may present challenges in Hgb measurement, they present a promising opportunity to use the app to screen for such diseases.

Minor comment:

-Supplementary Figure S4: number of anemia positive individuals should be 66 and not 77 in the Final diagnosis box

We thank the reviewer for the detailed examination of our manuscript. We have updated this figure in the revised manuscript.

Reviewer #3 (Remarks to the Author):

The manuscript by Mannino et al describes an imaging-based approach to measure blood hemoglobin (Hgb) levels. Specifically, the authors used a smartphone to take photos of fingernail beds, and developed customized image processing algorithms to estimate Hgb levels. The entire assay can be performed using the smartphone only, not requiring additional attachments or calibration tools. Results obtained by assessing the Hgb levels from the fingernail beds were quantitatively comparable to those obtained with a conventional complete blood count (CBC). Additional personalized calibration based on the CBC as a reference increases accuracy of the Hgb level analysis. While the current work presents a good level of technical advances, its major concepts are hardly new:

1. Using physical examination (color tint of the lower eyelid conjunctiva, nail-bed rubor, nail-bed blanching, and palmar crease rubor) to diagnose anemia has been known as shown in Ref. 17

We thank the reviewer for their careful review of our work and for highlighting the use of other noninvasive methodologies for diagnosing anemia. In rereading our manuscript, we realize that a lack of precision in our wording may have overstated the diagnostic capability and prevalence of physical examination. Only a few published studies have related pallor to anemia, and those provided only qualitative relationship rather than provide an estimation of Hgb levels. For example, the research group in ref. 17 characterized pallor in different body regions as a binary variable (i.e. whether or not a hematologist identifies clinical pallor in a patient). This method is insufficient to analyze diagnostic accuracy as it only establishes a link between anemia and pallor. In other words, as the binary presence of anemia was simply correlated with Hgb level, the only conclusion that can be drawn from this study is that pallor is linked to anemia. This approach is not sufficient to effectively diagnose anemia, in which an estimate of Hgb level is needed. To demonstrate this, we conducted an additional study where we show that clinical hematologists are unable to accurately measure Hgb levels via physical examination. Indeed, the 95% LOA of the Hgb level estimations of 5 trained hematologists (M.D. physicians who specialize in clinical hematology and are trained and Board Certified in the U.S.A.) was determined to be ± 4.6 g/dL of the CBC (n = 50), spanning nearly the entire range of tested Hgb. Receiver operating characteristic (ROC) analysis reveals an area under the curve of 0.63, indicating that physical examination by a hematologist performs barely better than a coin flip at diagnosing anemia. We have altered the introduction section to reflect the limited clinical use of physical examinations for characterizing anemia, and added an additional experiment and discussion to the manuscript.

Fig. 6: Smartphone app outperforms trained hematologists in hemaglobin measurement based on physical examination. Hematologists were able to estimate Hgb levels to within ± 4.6 g/dL (A) (95% limits of agreement) with an ROC of 0.63 (C). The app outperforms the hematologists in both respects with Hgb level accuracy measurement to within ± 1.0 g/dL (B) (95% limits of agreement) and an ROC of 0.94 (D), $n = 50$ subjects. Plots (A,C) represent the pooled results of 5 board certified hematologists estimating blood hemoglobin levels based on images of patients fingernails.

Main Text: Interestingly, several reports have suggested that anemia may qualitatively correlate with subjective assessments of pallor in various anatomic regions of the patient's body, namely the fingernail beds, conjunctiva, and pallor creases.

Results: The app outperforms clinical hematologists' ability to measure Hgb levels via physical examination.

Clinical hematologists, US Board certified physicians who specialize in the clinical care of patient with blood disorders, were asked to measure Hgb levels in patients via inspection of images of fingernails. In order to account for physician bias associated with physical examination of patients (e.g. prior knowledge of the patient's medical history), the physicians reviewed the same images of the patients' fingernails as the app. This approach better compares

the diagnostic capabilities of physicians and the app. When estimating Hgb levels based on examinations of images of patient fingernail beds ($n = 50$), hematologists estimated blood Hgb levels to within ± 4.6 g/dL of the CBC Hgb level (Fig 6A). Note that this degree of accuracy represents nearly the entire physiologic Hgb level range tested. The app was then tested on the dataset of 50 patient images and measured Hgb to within ± 1.0 g/dL of CBC Hgb levels (Fig 6B). Furthermore, ROC analysis revealed an area under the curve of 0.94 for the app vs 0.63 for the hematologists, representing a significant improvement in diagnostic accuracy (Fig 6C-D). Moreover, agreement of Hgb levels between the physicians' estimates, the smartphone app, and CBC Hgb levels was assessed using the intraclass correlation coefficient (ICC), which found that the smartphone all and CBC Hgb levels demonstrate excellent agreement (defined as $ICC < .9$) as ICC is estimated to be 0.95 (95% confidence interval (CI): 0.92 - 0.97), while an average of the 5 hematologists' evaluations demonstrated only moderate agreement with the CBC Hgb levels, with an ICC of 0.59 (0.37 - 0.74). Importantly, inter-physician variability of Hgb level estimates were high, as indicated by the low level of agreement with an ICC of 0.20 (95% CI .07 - 0.36).

Methods: Hemoglobin measurement from images of fingernails

Images were taken of 50 subjects fingernails from the previously described clinical study. These subjects' ages ranged from 1 to 62 years old. Hematologists (M.D. physicians who specialize in clinical hematology and are trained and Board Certified in the U.S.A.) were instructed to analyze each image and measure Hgb levels. For comparison, images were loaded into the app, and the Hgb measurement protocol was performed on these images. All images and analysis were taken using an iPhone 5S. It is important to note that these images were not used in the development of the underlying image analysis algorithm.

Intraclass correlation coefficient (ICC) reflects not only degree of correlation but also agreement between measurements and ranges between 0 and 1, with values closer to 1 representing stronger reliability. Reliability refers to the degree of agreement among raters. It gives a score of how much homogeneity, or consensus, there is in the ratings given by different judges or instruments. The ICC is able to incorporate the reliability of more than 2 raters-as in the case of the 5 hematologists evaluating nail beds. Patients and the physicians were assumed to be random samples from the respective populations they represent.

2. *Using a smartphone for point-of-care anemia diagnosis has been reported (Ref. 21 PLoS One 11:e0153286), which analyzed photos of conjunctival pallor. The achieved accuracy (AUC, 0.86) was similar to the current work (AUC, 0.87).*

We appreciate the reviewer's critiques and diligence in ensuring that Nature Communications continues to publish extremely novel technologies. In reviewing this comment, we feel that it is important to: 1) better contextualize reference 21, especially in regards to the present work and 2) better highlight the conceptual advance represented by this work.

1. **Contextualizing Ref 25** While the smartphone-based anemia measurement technique referenced by the reviewer (ref. 21) represents important work and measures Hgb based on images taken on a smartphone, there are a number of critical differences in our work that radically change how our approach will be used. First, the method described in reference 21 requires that all images be imported into a computer where a custom image analysis program was used to process the images and calculate Hgb. Second, self-testing with the approach described in ref 21 is logistically challenging due to the fact that taking

a photo of the conjunctiva requires the user to take an image at an awkward angle without being able to see the camera preview prior to taking the photo. Finally, the method described in ref 21 requires a color calibration card to be used within each image to account for variability between lighting conditions. This card must undergo rigorous quality control to ensure that each card manufactured possesses precisely the same colors, ensuring accurate color calibration. Furthermore, a user must acquire this card and use it properly, which is a non-trivial task for subjects measuring Hgb frequently. Taken together, these challenges represent a significant logistical hurdle to on-demand Hgb measurement in POC settings in which a user may not have easy access to a computer or a calibration card. We believe that our technology is ideally suited for these settings, as the lack of any additional equipment other than a smartphone allows patients to measure their Hgb on-demand from anywhere in the world.

While the diagnostic performance that this group has reported is similar to the accuracy that we report, their sample size is significantly less ($n = 47$ vs $n = 100$) with only 15 anemic patients. The AUC can be particularly variable with such a small sample size of anemic patients ($n = 15$ vs $n = 53$). Furthermore ref 21 does not report 95% LOA of the raw Hgb level results a key metric in evaluating the accuracy of a Hgb level measurement test. The lack of this vital diagnostic accuracy statistics significantly obfuscates accuracy comparisons with our technology.

2. **Conceptual Advance:** With regards to the conceptual novelty of the system, we agree with the reviewer that the manuscript text needed substantial changes to better highlight the conceptual advance of our technology. The reviewer has astutely pointed out a number of technical advances that we have made by developing a fully integrated and automated smartphone app that eliminates the need for a computer and specialized training. However, some of the technical advances that we have achieved such as the ease of use, fast processing speed (~ 30 sec) to obtain results, the lack of requiring any additional external equipment, and skin tone and background lighting agnostic Hgb level measurements, were only possible due to our significant, yet nuanced, conceptual advance. Specifically, our app technology leverages the image metadata, a vast trove of information that has been completely ignored by every published study to date that uses smartphones for diagnostics. By mining this rich source of information as well as the color data with a robust multi-linear regression approach, we demonstrate the first and only system to date in which accurate Hgb measurements are obtained with a smartphone without the need for any external equipment. Indeed, while smartphone images automatically record metadata, instead of examining this data, other groups have used physical strategies such as color calibration cards (ref 21) and light blocking enclosures (HemaApp). to compensate for the innate variability of color and light in smartphone images. This approach introduces a significant hurdle to the usability of these technologies, as they each require a user to acquire and properly use additional equipment.

We have summarized these key conceptual advances points in the discussion section of the manuscript below and better highlighted the substantial technological and conceptual advances represented by our work in a new Table 1 (see below):

Overall, the ability to conduct self-testing using an unmodified smartphone presents significant advantages over previously reported technologies which require additional equipment such as calibration cards and light-blocking rigs. Moreover, the app utilizes metadata that is

automatically obtained from the smartphone camera which enables normalization of background lighting conditions. This presents significant conceptual advantages over existing Hgb measurement technologies, as Hgb levels can now be measured by a patient without requiring a clinic visit or any cumbersome external equipment.

Device	Non-invasive	Smartphone-Based	No Additional Equipment (aside from smartphone)	Low-Cost	Accurate
Complete Blood Count (Gold Standard)	X	X	X	X	✓
Hemocue	X	X	X	X	✓
Masimo Co-Oximetry	✓	X	X	X	X
Conjunctival Analysis	✓	✓	X	✓	X
WHO Color Scale	X	X	X	✓	X
HemaApp	✓	✓	X	✓	X
Smartphone Anemia App	✓	✓	✓	✓	✓

Table 1: Comparison table of currently used anemia diagnostic technologies with the smartphone app

3. There are several FDA-approved POC devices (Masimo Pronto-7, OrSense NBM200) for non-invasive blood analyses. These devices not only check Hgb levels but also other indicators such SpO₂, perfusion, pulse rate. In the line of these assessments, the current work might be better suited for technical journals.

We value the reviewer’s feedback regarding the existence of FDA-approved devices that noninvasively monitor Hgb levels. While these absorbance-based systems are capable of monitoring Hgb levels noninvasively, they possess notable disadvantages. The primary disadvantage of these systems compared to the work we have presented is that these techniques require expensive instruments and depend on a clinician or technician with specialized training to operate. We have leveraged the fact that a large percentage of the population already owns and uses smartphones, to develop an inexpensive, noninvasive POC diagnostic for anemia that relies only on an unmodified smartphone, thus improving the accessibility of anemia screening and diagnosis over existing technologies. Furthermore, we have shown an improvement in anemia diagnostic accuracy over these technologies where our technology achieves an accuracy of ± 2.4 g/dL of the CBC while the Masimo Pronto-7 accuracy has been reported to be as much as ± 3.7 g/dL (ref 19 Moore *et al. American Journal of Surgery* 2013) of the CBC. As devices possessing

this level of accuracy (e.g. Hemocue 95% LOA = ± 2.3 g/dL of the CBC ref 19 Moore *et al. American Journal of Surgery* 2013) are widely used in POC settings, this level of accuracy clearly is ideal for screening in POC settings. This is important to note as, even with FDA approval, these devices come with disclaimers stating that these techniques should not be used in lieu of traditional blood-based testing to determine Hgb level and make clinical decisions and therefore, function as anemia screening tools, which is what our system will function as for the general population. Finally, while we have yet to explore such cases, the method we present has the potential to be applied to other disorders that manifest in physical discolorations (e.g. cyanosis, jaundice) which may provide an additional conceptual advance over these systems. We have included a comparison of our system's accuracy with these FDA-approved technologies in the revised manuscript. Additionally, we have discussed the conceptual advances of our system over these noninvasive techniques, namely the ability to conduct POC anemia diagnosis away from the clinic. Indeed, our technology is the only truly widely available approach in which anyone who wishes to monitor their anemia level simply downloads an app, takes a picture, and receives an accurate measurement.

Please refer to a new figure 1 below and table 1 above.

Fig. 1. Implementation of a smartphone image analysis system into a smartphone app enables non-invasive, patient-operated measurement of blood hemoglobin (Hgb) levels and

anemia detection using only patient-sourced photos and the native hardware of the smartphone itself. **A)** A patient simply downloads the app onto their smartphone, opens the app, obtains a smartphone photo of his/her fingernail beds, and without the need for any blood sampling or additional smartphone attachments or external calibration tools, quantitatively measures blood Hgb levels. The patient first takes an image of their fingernails, and is then prompted by the app to tap on the screen to select the regions of interest corresponding to the nailbeds, and a result is then displayed on the smartphone screen. Images are screenshots and photos of actual operation of this app. **B i)** As smartphone images with fingernail irregularities such as camera flash reflections or leukonychia may affect Hgb level measurements, a quality control algorithm integrated within the Hgb level measurement app detects and omits those irregularities to preserve measurement integrity and accuracy. **B ii)** To that end, the user selects regions of interest from within the fingernail image, **B iii)** and any color values that fall outside of expected color ranges are excluded from Hgb measurement. In this example, when the quality control system was implemented to exclude the fingernail bed irregularities, Hgb level was measured to be 14.7 g/dL, comparable to the patient's CBC Hgb level of 15.3 g/dL. Without the quality control algorithm, Hgb level was measured at 12.8 g/dL, indicating that the algorithm resulted in a 76% reduction in error. Note that as the smartphone image-based algorithm is device-agnostic, the analysis of the smartphone images, and therefore the Hgb level measurements, could also be transmitted to another device (e.g. laptop, cloud-based server) for remote rather than on-board analysis.

Main Text: Our approach represents a substantial conceptual advance over all other published POC anemia detection tools, since these techniques require external equipment, such as calibration cards, background light blocking devices, smartphone attachments, or expensive spectrometry readers. Here we have developed a fully functional and standalone smartphone app that enables the non-invasive measurement of blood Hgb levels, and has several advantages over existing approaches as highlighted in Table 1. Specifically, our app technology leverages the image metadata, a vast trove of information that has been completely ignored by every published study to date that uses smartphones for diagnostics. By mining this rich source of information as well as the color data with a robust multi-linear regression approach, we demonstrate the first and only system to date in which accurate Hgb measurements are obtained with a smartphone without the need for any external equipment. Indeed, while smartphone images automatically record metadata, instead of examining this data, other groups have used physical strategies such as color calibration cards and light blocking enclosures. This system enables "on demand" Hgb level measurement as it requires only the user's smartphone and no other equipment and can be conducted in under 1 minute. Therefore, users who desire to screen themselves for anemia can do so immediately by just downloading an app without being required to wait for external equipment to be shipped to their homes, which even other smartphone anemia tools require. Furthermore, this smartphone-based technique will empower patients to take control of their clinical care via self-testing of Hgb levels.

Discussion: Overall, the ability to conduct self-testing using an unmodified smartphone presents significant advantages over previously reported technologies which require additional equipment such as calibration cards and light-blocking rigs. Moreover, the app utilizes metadata that is automatically obtained from the smartphone camera which enables normalization of background lighting conditions. This presents significant conceptual advantages over existing Hgb measurement technologies, as Hgb levels can now be measured by a patient without requiring a clinic visit or any cumbersome external equipment.

Other minor comments:

- Please provide a reference to the Hgb level cutoff (12.5 g/dL) used in the manuscript. Official WHO criteria indicate different populations have different Hgb cutoff levels on anemia. The data set should be reorganized or additional data set is required to claim the capability of the system for the diagnosis of anemia considering the population and its corresponding Hgb level cutoff. What is the processing time of the photos to get the Hgb level?

We thank the review for taking the time to review our manuscript in such detail. This hemoglobin level cutoff was determined as the average Hgb level cutoff for a healthy female and a healthy male defined by the WHO in reference 6 (Worldwide prevalence of anemia 1993 – 2005: WHO 2008). We would also like to thank the reviewer in particular for inquiring about the time requirements of our test. The capability to rapidly measure Hgb levels is a key benefit of our system. It takes under 1 minute to measure Hgb level from the time a user opens the app to when a Hgb level result is displayed. This fact has been included in the manuscript.

This system enables “on demand” Hgb level measurement as it requires only the user’s smartphone and no other equipment and can be conducted in under 1 minute.

Reviewers' comments:

Reviewer #1 (Remarks to the Author):

Thank you: you have answered my comments

Reviewer #2 (Remarks to the Author):

The authors have addressed most of our previous comments. They have clarified the acceptability of the ± 2.6 g/dL Hgb detection accuracy, in the context of other currently available POC technology (e.g. Hemocue with ± 2.3 g/dL, Masimo Pronto 7 with ± 3.7 g/dL, WHO color scale at ± 3.3 g/dL etc). They have also shown improvement of performance to ± 2.4 g/dL using improved quality control algorithms. To better highlight the advantages of the Smartphone Anemia App in Table 1, the authors could modify the "Accurate" column to include what the acceptable range of accuracy is (alternatively list the accuracy of each method), as well as price to contextualize the "Low-Cost" column.

The authors have also added to the discussion on other medical conditions and diseases that could affect nailbed hue such as cyanosis, jaundice and hemolytic disorders. They have argued that these conditions are more significant than anemia, obviating the need for such patients to use the app under those circumstances. They have also acknowledged that additional studies are needed to study the impact of other disorders on Hgb measurement.

The authors have not commented on the accuracy statistic generated in the personal calibration use-case from a sample size of $N=4$ individuals, which seems insufficient for convincing statistical analysis of accuracy. With minor revisions addressing this sample size and Table 1, this manuscript is acceptable for publication in Nature Communication.

Minor Comment

- Figure 7 may not be necessary in the Main Text and could be moved to Supplementary (or even taken out completely).

Reviewer #4 (Remarks to the Author):

I was asked by the editor to give an opinion about whether the authors adequately responded to the critiques from Reviewer #3. After reviewing the manuscript, the reviewer comments, and the author's response, I believe that the authors thoroughly addressed the comments from Reviewer #3 in terms of novelty, comparison to other techniques, and the impact of the smartphone-based approach.

We would like to thank the reviewers for their careful consideration of our revised manuscript and their suggested edits to improve the overall clarity and impact of our work. We are particularly encouraged that the reviewers found our revised manuscript to be an improvement upon our originally submitted manuscript and that *Nature Communications* is “interested in the possibility of publishing [our] study”. We are further encouraged that Reviewer #1 and replacement Reviewer #3 believe that we have adequately addressed their comments and that Reviewer #2 believes that “with minor revisions addressing this sample size and Table 1, this manuscript is acceptable for publication”. However, Reviewer #2 brings up important concerns regarding the necessity of a discussion of sample size in the individual calibration of our hemoglobin level measurement algorithm, as well as the need to properly contextualize the accuracy and cost of our system relative to other alternatives in Table 1. Accordingly, we have addressed all of these concerns in a revised manuscript.

All changes to the revised manuscript are highlighted in yellow in this updated revised manuscript and in the responses below. Please see the response below for our point-by-point responses to the reviewer’s comments.

Reviewer #1 (Remarks to the Author):

Thank you: you have answered my comments

We thank the reviewer for the thorough review of our revised manuscript and we are pleased that we were able to adequately address the reviewer’s comments.

Reviewer #2 (Remarks to the Author):

The authors have addressed most of our previous comments. They have clarified the acceptability of the +/- 2.6g/dL Hgb detection accuracy, in the context of other currently available POC technology (e.g. Hemocue with +/- 2.3 g/dL, Masimo Pronto 7 with +/- 3.7g/dL, WHO color scale at +/- 3.3 g/dL etc). They have also shown improvement of performance to +/- 2.4g/dL using improved quality control algorithms. To better highlight the advantages of the Smartphone Anemia App in Table 1, the authors could modify the “Accurate” column to include what the acceptable range of accuracy is (alternatively list the accuracy of each method), as well as price to contextualize the “Low-Cost” column.

We would like to thank the reviewer for stating that we have “addressed most of [the reviewers] comments” and these for these new comments and very helpful suggestions. The reviewer has raised the excellent point that it is important to properly contextualize the price and accuracy of our technology. While these points are addressed in the text, they do not adequately come across in the summarizing Table 1 as intended. To that end, we have updated Table 1 to show price (< \$25) and accuracy (limits of agreement < 3.0 g/dL) levels of the different anemia diagnostic systems to better highlight and contextualized the advantages of the Smartphone Anemia App over the competing technologies.

The authors have also added to the discussion on other medical conditions and diseases that could affect nailbed hue such as cyanosis, jaundice and hemolytic disorders. They have argued

that these conditions are more significant than anemia, obviating the need for such patients to use the app under those circumstances. They have also acknowledged that additional studies are needed to study the impact of other disorders on Hgb measurement.

We are pleased that the reviewer found merit with our additional text that addressed the reviewer's previous concerns.

The authors have not commented on the accuracy statistic generated in the personal calibration use-case from a sample size of $N=4$ individuals, which seems insufficient for convincing statistical analysis of accuracy. With minor revisions addressing this sample size and Table 1, this manuscript is acceptable for publication in Nature Communication.

We thank the reviewer for bringing up point about how sample size reported in this experiment. Accordingly, we have addressed this comment by better clarifying how our statistical analysis is reported and have added text in both the results and discussion sections. In essence, our goal for these experiments is not to demonstrate that the personalized calibration distinguishes anemic patients from healthy controls, as this would not be possible for $N=4$ individuals as the reviewer pointed out, but rather that the personalized calibration yields consistent results for each individual and for that we have multiple data points for each individual. In addition, we state in the revised text that these results suggest that personalized calibration can further improve hemoglobin level measurement, but that this is also an ongoing area of research for us. Furthermore, these insightful comments led us to take a closer look into our reporting and justification of accuracy in the personalized calibration study.

In order to better characterize the agreement between two measurement methods with repeated hemoglobin levels for each subject, we have now used a modified Bland Altman analysis with intra-patient standard deviation (i.e. the standard deviation of measurement error of the 4 measurements within each of the 4 patients) estimated by a random effects model (new reference 27). The use of a random effects model to measure the 95% limits of agreement (LOA) is superior to what we reported previously due to the fact that it takes into account the dependent nature of the results (i.e. the 4 measurements of each individual taken over time are not independent). In addition to adjusting for intra-patient variance, this method can be used to adjust for other covariates (e.g. subject age, gender, etc.) and going forward, we will continue to collect those data. When conducting this updated analysis in combination with a thorough review of our existing analysis, we report an average error of ± 0.41 g/dL with 95% LOA of ± 0.92 g/dL and a bias of -0.09 g/dL. The 95% LOA ($1.96 * \text{Standard Deviation}$) is calculated using the standard deviation determined by the random effects model. Furthermore, 93% of measurements fall within Clinical Laboratory Improvement Amendment (CLIA) allowable total error of $\pm 7\%$. These results show that the average reported errors are consistent across the 4 patients, indicating a low inter-patient variability between the accuracy when tested on different patients (i.e. low error variance between patients). This is particularly notable given the diverse subject population, comprising two chronic anemia patients and two healthy controls spanning a ~ 6 g/dL Hgb range.

While this updated analysis led to small changes in some of our reported accuracy values (average error changed from 0.35 g/dL to 0.41 g/dL, accuracy changed from 0.90 g/dL to 0.92,

bias changed from 0.0 g/dL to 0.09 g/dL, and percentage of tests that fall within CLIA allowable total error for Hgb measurement improved from 88% to 93%), these changes were not sufficient to change our conclusions that the degree of accuracy reported by our system falls below a clinically significant threshold for Hgb level measurement of ± 1 g/dL, and that that, upon further refinement and completion of additional testing, this technology, when personally calibrated, may potentially be viable for at-home and clinical use for diagnosis of anemia.

Overall, as high variability skewing mean measurement error presents a key problem with reporting diagnostic test accuracy from datasets with small sample sizes, the low inter-patient variability we report as a result of our updated analysis (i.e. a random effects model which takes intra-patient variance into account) suggests that the reported accuracy values do not suffer from these issues and supports the conclusion that this app may be useful for self-measurement of Hgb levels given further study. To that end, the following text has been added to the manuscript:

Results: The standard deviation used to calculate the 95% LOA in this case was determined via a random effects model²⁷, which takes intra-patient variance caused by repeated Hgb measurements of each patient into account. This indicates that Hgb level measurement error is consistently low across our small, yet diverse study sample size (2 subjects suffering from chronic anemia, 1 healthy male subject, and 1 health female subject).

Furthermore, this degree of accuracy falls below a clinically significant threshold for Hgb level measurement of ± 1 g/dL,^{22,28,29} suggesting that this system can potentially be considered interchangeable with the CBC Hgb level given further study and an increased sample size³⁰

Discussion: Furthermore, the results of the app when individually calibrated suggest that this technology may, with further study, achieve Hgb measurement accuracy necessary for anemia diagnosis. Going forward, we will continue to increase enrollment in our individual calibration studies to confirm the high level of diagnostic accuracy that would be necessary to truly replace blood-based anemia testing.

Figure 4 Caption: A random effects model is used to statistically confirm consistency of average Hgb level measurement error between individual subjects.

Reference 27 describing the random effects model in the context of a Bland-Altman analysis has been added to the manuscript

Minor Comment:

- *Figure 7 may not be necessary in the Main Text and could be moved to Supplementary (or even taken out completely).*

We thank the reviewer for their assessment of our Figure 7. In accordance with the reviewer's suggestion, Figure 7 has been moved to the supplementary information as Figure S9.

REVIEWERS' COMMENTS:

Reviewer #2 (Remarks to the Author):

The authors have addressed our previous comments regarding the personal calibration use-case with a modified Bland Altman analysis and a random effects model to statistically confirm consistency of average Hgb level measurement error between individual subjects, as well as stating the need for future work with additional subjects to confirm these initial findings. This manuscript is suitable for publication in Nature Communications.

REVIEWERS' COMMENTS:

Reviewer #2 (Remarks to the Author): T

he authors have addressed our previous comments regarding the personal calibration use-case with a modified Bland Altman analysis and a random effects model to statistically confirm consistency of average Hgb level measurement error between individual subjects, as well as stating the need for future work with additional subjects to confirm these initial findings. This manuscript is suitable for publication in Nature Communications.

We would like to thank Reviewer #2 and all of the Reviewers for their thoughtful comments and helpful suggestions to improve the quality of our work. We are pleased that we were able to address all comments and that the Reviewers find our work suitable for publication in Nature Communications.